# DIMENSION REDUCTION AS AN OPTIMIZATION PROBLEM OVER A SET OF GENERALIZED FUNCTIONS

## ABSTRACT

We reformulate unsupervised dimension reduction problem (UDR) in the language of tempered distributions, i.e. as a problem of approximating an empirical probability density function $p_{\text{emp}}(\mathbf{x})$ by another tempered distribution $q(\mathbf{x})$ whose support is in a $k$-dimensional subspace. Thus, our problem is reduced to the minimization of the distance between $q$ and $p_{\text{emp}}$, $D(q, p_{\text{emp}})$, over a pertinent set of generalized functions.

This infinite-dimensional formulation allows to establish a connection with another classical problem of data science — the sufficient dimension reduction problem (SDR). Thus, an algorithm for the first problem induces an algorithm for the second and vice versa. In order to reduce an optimization problem over distributions to an optimization problem over ordinary functions we introduce a nonnegative penalty function $R(f)$ that "forces" the support of $f$ to be $k$-dimensional. Then we present an algorithm for minimization of $I(f) + \lambda R(f)$, based on the idea of two-step iterative computation, briefly described as a) an adaptation to real data and to fake data sampled around a $k$-dimensional subspace found at a previous iteration, b) calculation of a new $k$-dimensional subspace. We demonstrate the method on 4 examples (3 UDR and 1 SDR) using synthetic data and standard datasets.

## 1 INTRODUCTION

*Linear dimension reduction* (LDR) is a family of problems in data science that includes principal component analysis, factor analysis, linear multidimensional scaling, Fisher's linear discriminant analysis, canonical correlations analysis, sufficient dimensionality reduction (SDR), maximum autocorrelation factors, slow feature analysis and more. In unsupervised dimension reduction (UDR) we are given a finite number of points in $\mathbb{R}^n$ (sampled according to some unknown distribution) and the goal is to find a "low-dimensional" affine (or linear) subspace that approximates "the support" of the distribution. The study field currently achieved a saturation level at which unifying frameworks for the problem become of special interest Cunningham & Ghahramani (2015). An approach that we present in that paper is based on the theory of generalized functions, or tempered distributions Soboleff (1936); Schwartz (1949). An important generalized function that cannot be represented as an ordinary function is the Dirac delta function, denoted $\delta$, and $\delta^n$ denotes its $n$-dimensional version.

Any dataset $\{\mathbf{x}_i\}_{i=1}^N \subseteq \mathbb{R}^n$ naturally corresponds to the distribution $p_{\text{emp}}(\mathbf{x}) = \frac{1}{N} \sum_{i=1}^N \delta^n(\mathbf{x} - \mathbf{x}_i)$ which, with some abuse of terminology, can be called the empirical probability density function. Based on that, UDR can be understood as a task whose goal is to approximate $p_{\text{emp}}(\mathbf{x})$ by $q(\mathbf{x})$, where $q(\mathbf{x})$ is a distribution whose density is supported in a $k$-dimensional affine subspace $A \subseteq \mathbb{R}^n$. Note that a function whose density is supported in some low-dimensional subset of $\mathbb{R}^n$ is not an ordinary function. Exact definitions of such distributions can be found in Section 3. To formulate an optimization task we additionally need a loss $D(p_{\text{emp}}, q)$ that measures the distance between the ground truth $p_{\text{emp}}$ and a distribution $q$, that we search for. Thus, in our approach, the UDR problem is defined as:

$$I(q) = D(p_{\text{emp}}, q) \to \min_q \qquad (1)$$

under the condition that $q(\mathbf{x})$ has a $k$-dimensional support.

The SDR problem is tightly connected with the UDR problem. In SDR, given supervised data, the goal is to find the so called effective subspace, defined by its basis vectors $\{\mathbf{w}_1, \cdots, \mathbf{w}_k\} \subseteq \mathbb{R}^n$, such that the regression function can be searched in the form $g(\mathbf{w}_1^T \mathbf{x}, \cdots, \mathbf{w}_k^T \mathbf{x})$. In Wang et al. (2010) it was shown how a method originally developed for SDR can be turned into an UDR method, i.e. applied to unsupervised data, by simply setting an output to be equal to an input. The key observation of our analysis, stated in Theorem 2, is that a class of functions of the form $g(\mathbf{w}_1^T \mathbf{x}, \cdots, \mathbf{w}_k^T \mathbf{x})$ can be characterized as functions whose Fourier transform is supported in the corresponding effective subspace. In Section 4 we give 3 examples of UDR problems that we cast as 1 and in the fourth example we formulate SDR as an optimization task with the search space dual to that of UDR. Thus, all 4 examples can be studied within the same optimization framework.

The structure of the paper is as follows: in Section 3 we formally define the search space in Problem 1, denoted $\mathcal{G}_k$, and an image of $\mathcal{G}_k$ under the Fourier transform, denoted $\mathcal{F}_k$. Instead of searching directly in a set of generalized functions, $\mathcal{G}_k$, in Section 5 we describe how we substitute an ordinary function for a distribution in the optimization task at the expence of adding a new penalty term to its objective, $\lambda R(f)$. Using a gaussian kernel $M(\mathbf{x}, \mathbf{y})$, Theorem 4 characterizes generalized $g \in \mathcal{G}_k$ as such $g$ for which the matrix of properly defined integrals $M_g = \mathrm{Re} \left[ \iint_{\mathbb{R}^n \times \mathbb{R}^n} x_i y_j g(\mathbf{x})^* M(\mathbf{x}, \mathbf{y}) g(\mathbf{y}) d\mathbf{x} d\mathbf{y} \right]_{i,j=\overline{1,n}}$ is of rank $k$. We define $R(f)$ as a squared Frobenius distance from $\sqrt{M_f}$ to the closest matrix of rank $k$. In Section 6 we suggest a method for solving $\min_\phi I(\phi) + \lambda R(\phi)$ which we call *the alternating scheme*. Section 7 is dedicated to experiments with the alternating scheme on synthetic data and standard datasets.

## 2 PRELIMINARIES AND NOTATIONS

Throughout this paper we use standard terminology and notation from functional analysis. For exact definitions one can address the textbook on the theory of distributions Friedlander & Joshi (1998). The Schwartz space of functions and its dual space are denoted by $\mathcal{S}(\mathbb{R}^n)$ and $\mathcal{S}'(\mathbb{R}^n)$ correspondingly. For a tempered distribution $T \in \mathcal{S}'(\mathbb{R}^n)$ and $\phi \in \mathcal{S}(\mathbb{R}^n)$, $\langle T, \phi \rangle$ denotes $T(\phi)$. The Fourier and inverse Fourier transforms are denoted by $\mathcal{F}, \mathcal{F}^{-1} : \mathcal{S}'(\mathbb{R}^n) \to \mathcal{S}'(\mathbb{R}^n)$. For brevity, we denote $\mathcal{F}[f]$ by $\hat{f}$. If all required conditions are satisfied, an integrable $f : \mathbb{R}^n \to \mathbb{C}$ (or, a Borel measure $\mu$ on $\mathbb{R}^n$) is used as the tempered distribution $T_f$ (or, $T_\mu$) where $\langle T_f, \phi \rangle = \int_{\mathbb{R}^n} f(\mathbf{x}) \phi(\mathbf{x}) d\mathbf{x}$ (or, $\langle T_\mu, \phi \rangle = \int_{\mathbb{R}^n} \phi(\mathbf{x}) d\mu$). For $\Omega \subseteq \mathcal{S}'(\mathbb{R}^n)$, $\overline{\Omega}^*$ denotes the sequential closure of $\Omega$ with respect to weak topology of $\mathcal{S}'(\mathbb{R}^n)$. By $L_2(\mathbb{R}^n)$ we denote the $L_2$-space with the inner product: $\langle u, v \rangle_{L_2} = \int u(\mathbf{x})^* v(\mathbf{x}) d\mathbf{x}$. For $\phi \in \mathcal{S}(\mathbb{R}^n)$, $\psi \in \mathcal{S}'(\mathbb{R}^n)$, their convolution and multiplication are denoted by $\phi * \psi$ and $\phi\psi$ correspondingly. For $g_1 \in \mathcal{S}'(\mathbb{R}^k)$ and $g_2 \in \mathcal{S}'(\mathbb{R}^{n-k})$, $g_1 \otimes g_2 \in \mathcal{S}'(\mathbb{R}^n)$ denotes their tensor product. For a square matrix $A$, $\mathrm{Tr}(A)$ denotes its trace and for arbitrary matrix, $||A||_F \stackrel{def}{=} \sqrt{\mathrm{Tr}(A^T A)}$. Identity matrix of size $n$ is denoted by $I_n$.

## 3 BASIC FUNCTION CLASSES

An example of a generalized function, whose density is concentrated in a $k$-dimensional subspace, is any distribution that can be represented as $g \otimes \delta^{n-k} \stackrel{def}{=} g \otimes \underbrace{\delta \otimes \cdots \otimes \delta}_{n-k \text{ times}}$ where $g \in \mathcal{S}'(\mathbb{R}^k)$.

If $g = T_f$, where $f : \mathbb{R}^k \to \mathbb{R}$ is an ordinary function, then $g \otimes \delta^{n-k}$ can be understood as a generalized function whose density is concentrated in a subspace $\{\mathbf{x} \in \mathbb{R}^n | x_i = 0, i > k\}$ and equals $f(\mathbf{x}_{1:k})$. It can be shown that the distribution acts on $\phi \in \mathcal{S}(\mathbb{R}^n)$ in the following way:

$$\langle T_f \otimes \delta^{n-k}, \phi \rangle = \int_{\mathbb{R}^k} f(\mathbf{x}_{1:k}) \phi(\mathbf{x}_{1:k}, \mathbf{0}_{n-k}) d\mathbf{x}_{1:k}$$

Now to generalize the latter definition to any $k$-dimensional subspace we have to introduce a change of variables in tempered distributions.

Let $g \in \mathcal{S}'(\mathbb{R}^n)$ and $U \in \mathbb{R}^{n \times n}$ be an orthogonal matrix, i.e. $U^T U = I_n$. Then, $g_U \in \mathcal{S}'(\mathbb{R}^n)$ is defined by the rule: $\langle g_U, \phi \rangle = \langle g, \psi \rangle$ where $\psi(\mathbf{x}) = \phi(U^T \mathbf{x})$. If $g = T_f$, the latter definition gives $g_U = T_{f'}$ where $f'(\mathbf{x}) = f(U\mathbf{x})$. Now, we define classes of tempered distributions:

$$\mathcal{G}'_k = \{(f \otimes \delta^{n-k})_U | f \in \mathcal{S}'(\mathbb{R}^k), U \in \mathbb{R}^{n \times n}, U^T U = I_n\} \tag{2}$$

$$\mathcal{G}_k = \left\{ (T_f \otimes \delta^{n-k})_U | f \in \mathcal{S}(\mathbb{R}^k), U \in \mathbb{R}^{n \times n}, U^T U = I_n \right\} \tag{3}$$

$$\mathcal{F}_k = \{ T_r | r(\mathbf{x}) = f(U\mathbf{x}), f \in \mathcal{S}(\mathbb{R}^k), U \in \mathbb{R}^{k \times n}, \mathrm{rank}(U) = k \} \tag{4}$$

The first two classes are related as:

**Theorem 1.** $\mathcal{G}'_k = \overline{\mathcal{G}_k}^*$.

The last two classes are isomorphic under the Fourier transform.

**Theorem 2.** $\mathcal{F}[\mathcal{G}_k] = \mathcal{F}_k$ and $\mathcal{F}^{-1}[\mathcal{F}_k] = \mathcal{G}_k$.

For any collection $f_1, \cdots, f_l \in \mathcal{S}'(\mathbb{R}^n)$, $\mathrm{span}_{\mathbb{R}}\{f_i\}_1^l$ denotes $\{\sum_{i=1}^l \lambda_i f_i | \lambda_i \in \mathbb{R}\} \subseteq \mathcal{S}'(\mathbb{R}^n)$, which is a linear space over $\mathbb{R}$. The set $\mathcal{G}'_k$ has the following simple characterization:

**Theorem 3.** *For any $T \in \mathcal{S}'(\mathbb{R}^n)$, $T \in \mathcal{G}'_k$ if and only if $\dim \mathrm{span}_{\mathbb{R}}\{x_1 T, x_2 T, \cdots, x_n T\} \leq k$.*

*Informally*, the theorem holds because any linear dependency $\alpha_1 x_1 T + \cdots + \alpha_n x_n T = 0$ over $\mathbb{R}$ implies that if $\alpha_1 x_1 + \cdots + \alpha_n x_n \neq 0$, then $T = 0$. This is equivalent to a statement that the support of $T$ is concentrated on a subspace $\alpha_1 x_1 + \cdots + \alpha_n x_n = 0$. If $\dim \mathrm{span}_{\mathbb{R}}\{x_1 T, x_2 T, \cdots, x_n T\} \leq k$, then one can find $n - k$ such dependencies, which means that the support of $T$ is $k$-dimensional.

Let $\mathcal{B}(\mathbb{R}^n)$ denote the Borel sigma-algebra on $\mathbb{R}^n$ and $\mathcal{P}$ denote a set of all Borel probability measures on $\mathbb{R}^n$. Let us now define

$$\mathcal{P}_k = \{ \mu \in \mathcal{P} | \exists \mathbf{v}_1, \cdots, \mathbf{v}_k \in \mathbb{R}^n, \forall A \in \mathcal{B}(\mathbb{R}^n) : \mu(A) = \mu(A \cap \mathrm{span}(\mathbf{v}_1, \cdots, \mathbf{v}_k)) \} \tag{5}$$

i.e. $\mathcal{P}_k$ is a set of probability measures with all probability concentrated in some subspace $\mathrm{span}(\mathbf{v}_1, \cdots, \mathbf{v}_k)$ whose dimension is not greater than $k$. It is easy to see that $T_\mu \in \mathcal{G}'_k$ for any $\mu \in \mathcal{P}_k$.

## 4 EXAMPLES OF LDR FORMULATIONS

**UDR: Maximum mean discrepancy (MMD)** Let $k(\mathbf{x}) = \frac{1}{\sqrt{(2\pi h^2)^n}} e^{-\frac{|\mathbf{x}|^2}{2h^2}}$ be the radial gaussian kernel on $\mathbb{R}^n$. The kernel $k(\mathbf{x})$ defines the so-called kernel embedding of probability measures $\phi$ Muandet et al. (2017):

$$\mu \in \mathcal{P} \rightarrow \phi(\mu) = k * \mu = \mathbb{E}_{\mathbf{y} \sim \mu} k(\mathbf{x} - \mathbf{y}) = \int k(\mathbf{x} - \mathbf{y}) d\mu(\mathbf{y})$$

The Maximum Mean Discrepancy (MMD) distance Gretton et al. (2012) is defined as the distance induced by metrics on $L_2(\mathbb{R}^n)$, i.e. for two probability measures $\mu, \nu \in \mathcal{P}$:

$$d_{\mathrm{MMD}}(\mu, \nu) = ||\phi(\mu) - \phi(\nu)||_{L_2(\mathbb{R}^n)}$$

Let $\mathbf{x}_1, \cdots, \mathbf{x}_N \in \mathbb{R}^n$ be the dataset of points. This dataset defines the empirical probabilistic measure $\mu_{\mathrm{data}}$ that corresponds to the tempered distribution $T_{\mu_{\mathrm{data}}} = \frac{1}{N} \sum_{i=1}^N \delta^n(\mathbf{x} - \mathbf{x}_i)$. We shall study a method concurrent to PCA that is based on solving the following problem:

$$I(\nu) = d_{\mathrm{MMD}}(\mu_{\mathrm{data}}, \nu) = ||\phi(\mu_{\mathrm{data}}) - \phi(\nu)||_{L_2(\mathbb{R}^n)} \rightarrow \min_{\nu \in \mathcal{P}_k} \tag{6}$$

i.e. we shall attempt to approximate the empirical probabilistic measure $\mu_{\mathrm{data}}$ with another probabilistic measure $\nu$ which is supported in some $k$-dimensional subspace of $\mathbb{R}^n$.

**UDR: Distance based on higher moments (HM)** It is well-known that maximum mean discrepancy measures the similarity between characteristic functions of two probability distributions in the $O\left(\frac{1}{h}\right)$-neighbourhood of the origin. Another approach to measure the similarity of two distributions is based on the difference between moments:

$$d_{\mathrm{HM}}(\mu, \nu)^2 = \sum_{s=1}^4 \frac{\lambda_s}{n^s} \sum_{1 \leq i_1, \cdots, i_s \leq n} (m_{i_1 \cdots i_s} - n_{i_1 \cdots i_s})^2$$

where $m_{i_1 \cdots i_s} = \mathbb{E}_{\mathbf{X} \sim \mu}[\mathbf{X}[i_1] \cdots \mathbf{X}[i_s]]$ and $n_{i_1 \cdots i_s} = \mathbb{E}_{\mathbf{X} \sim \nu}[\mathbf{X}[i_1] \cdots \mathbf{X}[i_s]]$ are corresponding moments. The positive parameters $\lambda_1, \lambda_2, \lambda_3, \lambda_4$ are chosen to fix the relative importance of the mean, the co-variance, the co-skewness and the co-kurtosis.

Thus, we will be interested in the following optimization task (analogous to 6):

$$d_{\text{HM}}(\mu_{\text{data}}, \nu) \to \min_{\nu \in \mathcal{P}_k} \tag{7}$$

**UDR: Wasserstein distance (WD)** Another important distance between probability measures that has the origins in the transport theory is the Wasserstein distance Villani (2008).

Let $(\mathbb{R}^n, || \cdot ||)$ be a Banach space. Between any two Borel probability measures $\mu, \nu$ on $\mathbb{R}^n$ with $\int ||\mathbf{x}|| d\mu < \infty$ and $\int ||\mathbf{x}|| d\nu < \infty$ the Wasserstein distance is:

$$W(\mu, \nu) = \inf_{\pi \in \Pi(\mu, \nu)} \int ||\mathbf{x} - \mathbf{y}|| d\pi$$

where $\Pi(\mu, \nu)$ is a set of all couplings of $\mu$ and $\nu$. The Wasserstein distance defines another version of LDR problem:

$$W(\mu_{\text{data}}, \nu) \to \min_{\nu \in \mathcal{P}_k} \tag{8}$$

In the appendix B one can find proofs that in the case of $L_1$ norm $||\mathbf{x}|| = \sum_i |x_i|$, the task 8 corresponds to the well-studied *robust PCA* problem Candès et al. (2011). If, instead of the $L_1$-norm, we use the $L_2$-norm, this leads to another well-studied task, which is known as *the outlier pursuit* problem Xu et al. (2010).

**Sufficient dimension reduction (SDR)** Given a labeled dataset $\{(\mathbf{x}_i, y_i)\}_{i=1}^N$ where $\mathbf{x}_i \in \mathbb{R}^n, y_i \in \mathcal{C}$ ($\mathcal{C}$ is a finite set of classes for a classification, or $\mathbb{R}$ for a regression problem), the sufficient dimension reduction problem can be informally described as a problem of finding vectors $\mathbf{w}_1, \cdots, \mathbf{w}_k \in \mathbb{R}^n$ such that $p(y|\mathbf{w}_1^T\mathbf{x}, \cdots, \mathbf{w}_k^T\mathbf{x}) \approx p(y|\mathbf{x})$ (possibly, under some additional assumptions on the form of $p(y|\mathbf{x})$).

We formulate the SDR problem as an optimization task:

$$\inf_{f \in \mathcal{F}_k} J(f) \tag{9}$$

The object $f : \mathbb{R}^n \to \mathbb{R}$ is a smooth real-valued function. We assume that $f$ is a candidate for the regression function and $J(f)$ is a cost function that values how strongly $f$ fits in this role. In practice for the regression case and for the binary classification case with 0-1 outputs we use the following cost functions correspondingly:

$$J(f) = \frac{1}{N} \sum_{i=1}^N \mathbb{E}_{\boldsymbol{\epsilon} \sim N(\mathbf{0}, \upsilon^2 I_n)} |y_i - f(\mathbf{x}_i + \boldsymbol{\epsilon})|^2$$

$$J(f) = \frac{1}{N} \sum_{i=1}^N \mathbb{E}_{\boldsymbol{\epsilon} \sim N(\mathbf{0}, \upsilon^2 I_n)} H\left(y_i, \frac{e^{f(\mathbf{x}_i + \boldsymbol{\epsilon})}}{1 + e^{f(\mathbf{x}_i + \boldsymbol{\epsilon})}}\right)$$

where $H(y, p) = -y \log p - (1 - y) \log(1 - p)$ and $\upsilon > 0$ is a parameter.

By requiring $f \in \mathcal{F}_k$, we assume that the regression function $f$ satisfies (for $k$ fixed in advance):

$$f(\mathbf{x}) = g(\mathbf{w}_1^T\mathbf{x}, \cdots, \mathbf{w}_k^T\mathbf{x})$$

where $\mathbf{w}_1, \cdots, \mathbf{w}_k \in \mathbb{R}^n$. Thus, given an input $\mathbf{x}$, an output of $f$ depends on the projection of $\mathbf{x}$ onto $\text{span}(\mathbf{w}_1, \cdots, \mathbf{w}_k)$. The set $\text{span}(\mathbf{w}_1, \cdots, \mathbf{w}_k)$ is called the effective subspace. Note that the way we defined the SDR's objective $J(f)$ for the regression and the classification cases is not unique. There are definitions that has the same form 9, but deal with the conditional distribution $p(y|\mathbf{x})$ as an argument, instead of the regression function.

## 5 REDUCTION OF THE OPTIMIZATION PROBLEM TO ORDINARY FUNCTIONS

The central problem that our paper addresses is how to minimize an objective function over $\mathcal{G}'_k$ (or $\mathcal{P}_k$)? In this section we describe an approach based on penalty functions and kernels.

Let us assume for simplicity that $M$ is the gaussian kernel, i.e. $M(\mathbf{x}, \mathbf{y}) = G_\sigma^n(\mathbf{x} - \mathbf{y})$ where $G_\sigma^n(\mathbf{x}) = \frac{1}{\sqrt{2\pi\sigma^2}^n} e^{-\frac{|\mathbf{x}|^2}{2\sigma^2}}$. Besides the gaussian kernel our theory also captures many other kernels,

including cases of the Abel Kernel $\frac{1}{\sigma^n} e^{-\frac{|\mathbf{x}-\mathbf{y}|}{\sigma}}$ and the Fourier tranform of the Abel Kernel, the Poisson kernel: $\frac{c_n \sigma}{(\sigma^2 + |\mathbf{x}-\mathbf{y}|^2)^{\frac{n+1}{2}}}$.

For $f, g \in \mathcal{S}(\mathbb{R}^n)$ let us denote:

$$\langle f|M|g\rangle \overset{def}{=} \iint_{\mathbb{R}^n \times \mathbb{R}^n} f(\mathbf{x})^* M(\mathbf{x}, \mathbf{y}) g(\mathbf{y}) d\mathbf{x} d\mathbf{y} \leq \max_{\mathbf{x}, \mathbf{y}} M(\mathbf{x}, \mathbf{y}) \|f\|_{L_1} \|g\|_{L_1} < \infty$$

For general $f, g \in \mathcal{S}'(\mathbb{R}^n)$ the expression $\langle f|M|g\rangle$ is defined if $\exists f_\epsilon, g_\epsilon \in \mathcal{S}(\mathbb{R}^n)$ such that $T_{f_\epsilon} = f * G_\epsilon^n$, $T_{g_\epsilon} = g * G_\epsilon^n$ and $\langle f_\epsilon|M|g_\epsilon\rangle \overset{\epsilon \to 0}{\to} A$. Then, $\langle f|M|g\rangle \overset{def}{=} A$. For example, $\langle \delta^n|M|\delta^n\rangle = M(0,0)$.

Theorem 3 concludes, from $f \in \mathcal{G}_k$, that $\dim \mathrm{span}_{\mathbb{R}}\{x_1 f, x_2 f, \cdots, x_n f\} \leq k$. Using the kernel $M$, one can build the Gram matrix from the collection of distributions, $[\langle x_i f|M|x_j f\rangle]_{1 \leq i,j \leq n}$. For any $f \in \mathcal{S}'(\mathbb{R}^n)$ let us denote a real part of the Gram matrix $[\mathrm{Re}\,\langle x_i f|M|x_j f\rangle]_{1 \leq i,j \leq n}$ by $M_f$ (if it is defined).

**Theorem 4.** *If $f \in \mathcal{G}_k$, then $\langle x_i f|M|x_j f\rangle$ is defined and $\mathrm{rank}\, M_f \leq k$.*

**Definition 1.** *Let $A \in \mathbb{R}^{n \times n}$ be a positive semidefinite matrix with eigenvalues $\lambda_1 \geq \lambda_2 \geq \cdots \geq \lambda_n$ (with counting multiplicities). Then, the Ky Fan $k$–anti-norm of $A$ is $\|A\|_k = \sum_{i=1}^{k} \lambda_{n+1-k}$.*

Let $R(f) = \|M_f\|_{n-k}$. Theorem 4 tells us that that for $f \in \mathcal{G}_k$, $R(f) = 0$. For ordinary $f$, the Eckart-Young-Mirsky theorem gives us $R(f) = \min_{A \in \mathbb{R}^{n \times n}, \mathrm{rank}\, A \leq k} \|\sqrt{M_f} - A\|_F^2$. Thus, by penalizing the value of $R(f)$, we enforce $M_f$ to be close to some matrix of rank $k$. For $I : \mathcal{G}_k' \cup \mathcal{S}(\mathbb{R}^n) \to \mathbb{R}^+$, it is natural to reduce the optimization task *over tempered distributions*

$$I(f) \to \min_{f \in \mathcal{G}_k'} \tag{10}$$

to an optimization task *over ordinary functions with a penalty term $R$*:

$$I(f) + \lambda \|M_f\|_{n-k} = I(f) + \lambda R(f) \to \inf_{f \in \mathcal{S}(\mathbb{R}^n)} \tag{11}$$

Details on the conditions, under which this reduction holds, can be found in the appendix D. Let us now concentrate on the task 11 and describe the alternating scheme for its solution.

## 6 THE ALTERNATING SCHEME

We will concentrate on problem 11. It is known Hiai (2013) that the Ky Fan anti-norm is a concave function, i.e. $R(\phi) = \|M_\phi\|_{n-k}$ depends on $M_\phi$ in a concave way. It can be shown that the dependence of $R(\phi)$ on $\phi$ is both non-convex and non-concave, i.e. we deal with a non-convex optimization task.

The kernel $M(\mathbf{x}, \mathbf{y}) : \mathbb{R}^n \times \mathbb{R}^n \to \mathbb{C}$ induces a linear operator from $L_2(\mathbb{R}^n)$ to $L_2(\mathbb{R}^n)$: $O(M)[f] = \int_{\mathbb{R}^n} M(\mathbf{x}, \mathbf{y}) f(\mathbf{y}) d\mathbf{y}$. For any operator $O$ between spaces $\mathcal{H}_1$ and $\mathcal{H}_2$, we denote its range as $\mathcal{R}[O] = \{O(x)|x \in \mathcal{H}_1\}$. Let $\mathcal{B}(H_1, H_2)$ denote a set of bounded linear operators between Hilbert spaces $H_1$ and $H_2$. For $O \in \mathcal{B}(H_1, H_2)$ the rank of $O$ is defined as $\dim \mathcal{R}(O)$. Let $L_2^r(\mathbb{R}^n)$ be the Hilbert space (over $\mathbb{R}$) of real-valued functions from $L_2(\mathbb{R}^n)$ and $L_2^*(\mathbb{R}^n) = L_2^r(\mathbb{R}^n) \times L_2^r(\mathbb{R}^n)$. The space $L_2^*(\mathbb{R}^n)$ is equivalent to $L_2(\mathbb{R}^n)$ treated as a linear space over $\mathbb{R}$. Below we do not distinguish $[\phi_1, \phi_2] \in L_2^*(\mathbb{R}^n)$ and $\phi_1 + i\phi_2 \in L_2(\mathbb{R}^n)$. It is easy to see that any $O \in \mathcal{B}(L_2^*(\mathbb{R}^n), \mathbb{R}^n)$ can be given by formula:

$$O[\phi]_i = \mathrm{Re}\,\langle O_i, \phi\rangle_{L_2(\mathbb{R}^n)}, O_i \in L_2(\mathbb{R}^n), i = \overline{1, n}$$

i.e. $O \in \mathcal{B}(L_2^*(\mathbb{R}^n), \mathbb{R}^n)$ can be identified with a vector of functions $O = [O_i]_{i=\overline{1,n}}, O_i \in L_2(\mathbb{R}^n)$ and the Hilbert–Schmidt norm on $\mathcal{B}(L_2^*(\mathbb{R}^n), \mathbb{R}^n)$ (i.e. $\sqrt{\mathrm{Tr}\, O^\dagger O}$) is:

$$\|O\|_* = \sqrt{\sum_{i=1}^{n} \|O_i\|_{L_2(\mathbb{R}^n)}^2} \tag{12}$$

Recall that for the kernel $M$, $O(M)$ is positive and self-adjoint. Since $O(M)$ is also bounded, then the square root $\sqrt{O(M)}$ can be correctly defined Rudin (1991). For any complex-valued function $f$ let us introduce a linear operator $S_f : L_2^*(\mathbb{R}^n) \to \mathbb{R}^n$ by the following rule:

$$S_f[\phi]_i = \text{Re} \langle x_i f(\mathbf{x}), \sqrt{O(M)}[\phi] \rangle_{L_2(\mathbb{R}^n)} \text{ i.e. } (S_f)_i = \sqrt{O(M)}[x_i f(\mathbf{x})], i = \overline{1,n}$$

**Theorem 5.** *If $\text{Tr}\, M_f < \infty$, then $S_f \in \mathcal{B}(L_2^*(\mathbb{R}^n), \mathbb{R}^n)$ and $S_f S_f^\dagger = M_f$. Moreover,*

$$R(f) = \min_{S \in \mathcal{B}(L_2^*(\mathbb{R}^n), \mathbb{R}^n), \text{rank}\, S \leq k} ||S_f - S||_*^2$$

*and the minimum is attained at $S = P_f S_f$ where $P_f = \sum_{i=1}^k \mathbf{u}_i \mathbf{u}_i^\dagger$ and $\{\mathbf{u}_i\}_1^k$ are unit eigenvectors of $M_f$ corresponding to the $k$ largest eigenvalues (counting multiplicities).*

Given the new representation $R(f) = \min_{S \in \mathcal{B}(L_2^*(\mathbb{R}^n), \mathbb{R}^n), \text{rank}\, S \leq k} ||S_f - S||_*^2$ it is natural to view the Task 11 as a minimization of $I(\phi) + \lambda||S_\phi - S||_*^2$ over two objects: $\phi$ and $S \in \mathcal{B}(L_2^*(\mathbb{R}^n), \mathbb{R}^n)$ : $\text{rank}\, S \leq k$. The simplest approach to minimize a function over two arguments is to optimize alternatingly, i.e. first over $\phi$, and then over $S : \text{rank}\, S \leq k$, and so on. Theorem 5 gives that the minimization over $S$ is equivalent to the truncation of $\text{SVD}(S_\phi)$ at the $k$-th term. This idea, that we dub the *alternating scheme*, is described in Algorithm 1.

---

**Algorithm 1** The alternating scheme for 11

$P_0 \longleftarrow \mathbf{0}, S_{\phi_0} \longleftarrow \mathbf{0}$
**for** $t = 1, \cdots, T$ **do**
$\quad \phi_t \longleftarrow \arg\min_\phi I(\phi) + \lambda||S_\phi - P_{t-1} S_{\phi_{t-1}}||_*^2$ (minimizing over $\phi$)
$\quad$ Calculate $M_{\phi_t}$ and find $\{\mathbf{v}_i\}_1^n$ s.t. $M_{\phi_t}\mathbf{v}_i = \lambda_i \mathbf{v}_i, \lambda_1 \geq \cdots \geq \lambda_n$
$\quad P_t \longleftarrow \sum_{i=1}^k \mathbf{v}_i \mathbf{v}_i^T$ (Truncated $\text{SVD}(S_{\phi_t})$ is $P_t S_{\phi_t}$)
**Output:** $\mathbf{v}_1, \cdots, \mathbf{v}_k$

---

The alternating scheme 1 allows for a reformulation in the dual space. By this we mean that in Scheme 1 we substitute $\widehat{\phi}_t$ for the original $\phi_t$. If the primal Scheme 1 deals with operators $S_\phi, S_{\phi_{t-1}}$, the dual version deals with vectors of functions $\sqrt{\widehat{G}_\sigma} \frac{\partial \widehat{\phi}}{\partial \mathbf{x}}, \sqrt{\widehat{G}_\sigma} \frac{\partial \widehat{\phi}_{t-1}}{\partial \mathbf{x}}$. Details of the dual algorithm can be found in the appendix F.

## 7 EXPERIMENTS

The alternating scheme 1 is a general optimization method which needs to be specified for every optimization task. We designed numerical specifications of the alternating scheme 1 for all 4 optimization tasks: 6, 7, 8 and 9 and made experiments with all of them. Details of the algorithms, i.e. numerical methods to minimize over $\phi$ and calculate $M_{\phi_t}$, can be found in the appendix (G, H, I and J). Note that for Wasserstein distance minimization 8 we exploit the alternating scheme in the initial form (i.e. 1), and for MMD 6, HM 7 and SDR 9 we use the dual version of the scheme.

**Behaviour of MMD for small $h$.** We studied the difference in the behaviour of PCA and a solution of 6 obtained by the alternating scheme 1 (MMD), for the case when $h$ is small compared to the standard deviation of features. Experiments show that they are sharply different when data points are sampled along a low-dimensional manifold $\mathfrak{M}$, which is bent globally, goes through the origin $O$ and has a large curvature at $O$. Because PCA is a global method and points do not lie on an affine subspace, interpreting principal directions is not straightforward.

We select a smooth function $f : \mathbb{R}^{n-1} \to \mathbb{R}$, such that $f(\mathbf{0}) = 0$ and generate points in the following way: points $\mathbf{x}_1, \mathbf{x}_2, \cdots, \mathbf{x}_N \sim [-10, 10]^{n-1}$ are sampled uniformly, after calculation of $y_i = f(\mathbf{x}_i)$ we add some noise: $\mathbf{z}_i = (\mathbf{x}_i, y_i) + \boldsymbol{\epsilon}_i, \boldsymbol{\epsilon}_i \sim \mathcal{N}(0, 0.01 I_n)$. Both PCA and MMD are applied to the dataset (first 3 pictures on Figure 1a). As we see, MMD, unlike PCA, tries to catch ideal alignments of points rather that searching for a global alignment of points (which can be non-existent). This property of MMD makes it a promising tool for the calculation of the tangent space

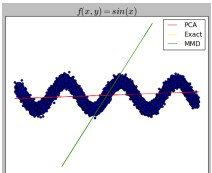 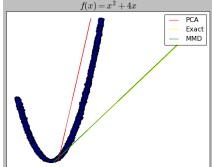 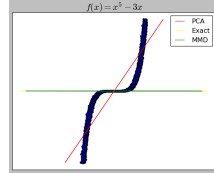 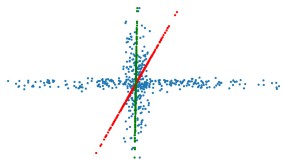

(a) Visualization of outputs of the PCA and MMD methods. MMD (green line) tends to select a subcollection of points that sharply aligns along the main direction, whereas the first principal component (red line) could be a result of averaging over different directions in the data.

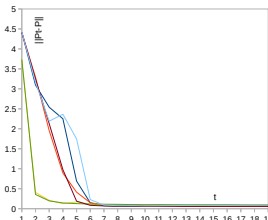 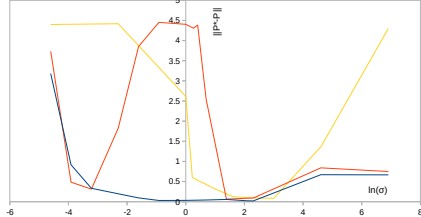

(b) Left plot: $||P_t - P||_F$: ■ $\delta = 0.05, \lambda = 20.0$, case I, ■ $\delta = 0.05, \lambda = 20.0$, case II, ■ $\delta = 0.05, \lambda = 100.0$, case I, ■ $\delta = 0.05, \lambda = 100.0$, case II, ■ $\delta = 0.1, \lambda = 100.0$, case I, ■ $\delta = 0.1, \lambda = 100.0$, case II. Right plot: $||P^* - P||_F$ as a function of $\ln \sigma$:■ MMD, ■ HM, ■ WD.

to a data manifold at a given point. Fourth picture shows that when we have 2 equally important directions in data such that the first principal direction of PCA is between them (red line), and we set $k = 1$, then MMD (green line) always chooses one of those directions.

**Experiments with outlier detection (MMD, HM, Wasserstein distance).** Following the experiment setup of Xu et al. (2010), we choose parameters $N = n = 400, \delta = 0.05(0.1), k = 10$ and generate random matrices $A \in \mathbb{R}^{N(1-\delta) \times k}, B \in \mathbb{R}^{n \times k}$ whose entries are iid as $\mathcal{N}(0, 1)$. Then, to the columns of the matrix $BA^T \in \mathbb{R}^{n \times N(1-\delta)}$ (whose rank is $\leq k$) are concatenated with the columns of the matrix $C \in \mathbb{R}^{n \times N\delta}$: $X = \text{concat}(BA^T, C) \in \mathbb{R}^{n \times N}$. The entries in $C$ are either iid as $\mathcal{N}(0, 1)$ (case I) or $N\delta$ copies of the same vector whose entries are iid as $\mathcal{N}(0, 1)$ (case II). Let $X = [\mathbf{x}_1, \cdots, \mathbf{x}_N]$, i.e. columns of $X$ are the data points. Thus, $N(1 - \delta)$ columns of $BA^T$ lie in a $k$-dimensional subspace of $\mathbb{R}^n$ and $N\delta$ columns of $C$ are outliers, and solutions of tasks 6, 7 or 8 for this dataset are expected to be supported in a column space of $BA^T$.

After every iteration (step $t$ of the alternating scheme 1) we calculate the Frobenius distance between the projection operator $P_t$ of 1 and the projection operator $P$ to the column space of $BA^T$, i.e. $||P_t - P||_F$. For the task 8, the dependence of $||P_t - P||_F$ on $t$ for different values of parameters $\delta$ and $\lambda$ is shown in Figure 1b. For tasks 6, 7 the behaviour of the alternating scheme is similar, 7 iterations are enough to approach the optimal subspace.

Besides the speed of convergence we were also interested in how $||P^* - P||_F$, where $P^* = \lim_{t \to \infty} P_t$ is the final projection operator (e.g. $P_{20}$ in practice), depends on the parameter $\sigma$ of the kernel $M = G_\sigma$. It is natural to expect the quality of the solution $P^*$ to degrade as $\sigma \to +\infty$ (this corresponds to $M(\mathbf{x}, \mathbf{y}) \to 0$), and, less trivially, as $\sigma \to 0$ (this corresponds to $M(\mathbf{x}, \mathbf{y}) \to \delta^n(\mathbf{x} - \mathbf{y})$).

**Experiments with the sufficient dimension reduction.** We made experiments on the standard datasets, Heart, Breast Cancer, Ionosphere, Diabetes, Boston house prices and Wine quality. First we applied Sliced Inverse Regression algorithm (SIR) Li (1991) to the training set and calculated the effective subspace for $k = 2, 3$. All points were projected onto that space and we obtained two- or three-dimensional representations of input points. In the last step we applied 10 nearest neighbours algorithm (KNN) to predict outputs (based on reduced inputs) on the test set (for the regression case, the 10-KNN regression was used). The same scheme was repeated with PCA, Kernel Dimensionality Reduction (KDR) algorithm Fukumizu et al. (2004) and the alternating scheme 1 adapted for SDR.

We experimented with the dual version of algorithm 1, setting (after the data was standardized) the kernel's parameter $\sigma = 0.8$[1] and $\lambda = 10.0$. Details of its numerical implementation can be found in the appendix J. In the table 1 one can see the obtained test set accuracy on the classification tasks and $R^2$ on the regression tasks. As we see from the table 1, after reducing the dimension of an input to $k = 2, 3$, we are still able to obtain good accuracy of prediction on a test set.

| Method / Dataset | PCA | | SIR | | KDR | | AS 1 | |
|---|---|---|---|---|---|---|---|---|
| Dimension $k$ | 2 | 3 | 2 | 3 | 2 | 3 | 2 | 3 |
| Heart (acc) | 79.80 | 79.46 | 82.49 | 81.82 | **86.33** | **88.77** | 81.48 | 83.50 |
| Breast (acc) | 93.46 | 93.65 | 97.30 | 96.73 | 93.13 | 95.95 | **97.88** | **97.69** |
| Ionosphere (acc) | 80.29 | 86.57 | **89.14** | 89.43 | 83.43 | 86.29 | 88.29 | **90.57** |
| Diabetes ($R^2$) | 25.34 | 28.72 | **43.47** | 43.61 | 41.82 | 44.30 | 43.07 | **44.48** |
| Boston ($R^2$) | 56.42 | 67.12 | 76.03 | 74.29 | **77.88** | **79.97** | 73.21 | 77.88 |
| Wine ($R^2$) | 93.91 | 94.12 | **98.68** | **99.24** | 98.30 | 96.02 | 97.10 | 96.93 |

Table 1: The cross-validated accuracies/$R^2$ of KNN on 2 or 3-dimensional input representations.

The code is available on github to facilitate the reproducibility of our results.

## 8 RELATED WORK

We present an optimization framework in which the search space is $\mathcal{G}'_k$, or $\mathcal{P}_k$. Another unifying framework for LDR tasks is suggested by Cunningham & Ghahramani (2015) in which the basic search space is the Stiefel manifold $\mathcal{S}(n, k)$. The main disadvantage of using $\mathcal{G}'_k$, instead of the Stiefel manifold, is that its infinite number of dimensions requires a special procedure to turn an optimization into a finite-dimensional task. Both an optimization over $\mathcal{G}'_k$ and over $\mathcal{S}(n, k)$ is typically hard: for a final point, at best one can guarantee that it is a local extremum. Promising aspects of $\mathcal{G}'_k$ are: a) $\mathcal{G}'_k$ allows to formulate naturally a new class of objectives on it, b) local extrema on $\mathcal{G}'_k$ substantially differ from local extrema on $\mathcal{S}(n, k)$, because a local search over $\mathcal{G}'_k$ uses more degrees of freedom.

Using Ky-Fan $k$-antinorm as a regularizer for the matrix completion problem has been suggested by Hu et al. (2013) and further developed in Oh et al. (2016); Liu et al. (2016); Hong et al. (2016). Unlike this chain of works, we formulate an infinite-dimensional task. Also, our regularizer $R(f) = ||M_f||_{n-k}$ is a sum of smallest $n-k$ squared singular values of the operator $S_f$ where $S_f$ depends on $f$ linearly. The idea of alternating two basic stages, the convex optimization and SVD, is ubiquitous in low-rank optimization, see e.g. Mazumder et al. (2010); Hastie et al. (2015).

Zhu & Zeng (2006) applied the Fourier transform for estimating the effective subspace in SDR, implicitly using an analog of Theorem 2.

## 9 CONCLUSIONS

We develop a new optimization framework for LDR problems. The alternating scheme for the optimization task demonstrates both the computational efficiency and the applicability to real-world data. The algorithm performs quite stably when we vary most of the hyperparameters, though it crucially depends on two parameters, the bandwidth of the "smoothing" kernel $M$, $\sigma$, and the penalty parameter $\lambda$. We believe that the MMD/HM/WD methods for UDR could be used as an alternative to PCA in study fields in which data demonstrate "heavy-tailed" and "non-gaussian" behaviour, such as financial applications. Also, our formulation of SDR is free from any assumptions on the distribution of input-output pairs, which makes it an alternative to other methods of the efficient subspace estimation. More detailed report on these topics is a subject of future research.

---

[1]Since the role of the parameter $\sigma$ is similar to that of the bandwidth in the kernel density estimation, we use Silverman's rule of thumb to set $\sigma = N^{-1/(n+4)}$.

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

## A PROOFS FOR SECTION 3

### A.1 PROOF OF THEOREM 1: GIVEN FOR COMPLETENESS

*Proof.* The inclusion $\mathcal{G}'_k \subseteq \overline{\mathcal{G}_k}^*$ follows from a well-known fact that $\mathcal{S}(\mathbb{R}^k)$ is dense in $\mathcal{S}'(\mathbb{R}^k)$. I.e. for any $f \in \mathcal{S}'(\mathbb{R}^k)$ one can always find a sequence $\{f_i\} \subseteq \mathcal{S}(\mathbb{R}^k)$ such that $T_{f_i} \to^* f$. Therefore, for any $(f \otimes \delta^{n-k})_U \in \mathcal{G}'_k$ there is a sequence $\{(T_{f_i} \otimes \delta^{n-k})_U\} \subseteq \mathcal{G}_k$ such that $(T_{f_i} \otimes \delta^{n-k})_U \to^* (f \otimes \delta^{n-k})_U$. Thus, $\mathcal{G}'_k \subseteq \overline{\mathcal{G}_k}^*$.

Since $\mathcal{G}_k \subseteq \mathcal{G}_k'$, to prove $\mathcal{G}_k' = \overline{\mathcal{G}_k}^*$ it is enough to show that $\mathcal{G}_k'$ is sequentially closed.

We need a simple fact from a theory of distributions.

**Lemma 1.** *If $T_i \to^* T$ and $\phi_i \to \phi$, then $\langle T_i, \phi_i \rangle \to \langle T, \phi \rangle$.*

*Proof of Lemma.* Schwartz space $\mathcal{S}(\mathbb{R}^n)$ is a Fréchet space, therefore the Banach-Steinhaus theorem applies to $\mathcal{S}'(\mathbb{R}^n)$. Since $T_i \to^* T$, we have $\sup_i |\langle T_i, \phi \rangle| < \infty$ for any $\phi \in \mathcal{S}(\mathbb{R}^n)$. From the Banach-Steinhaus theorem, applied to a set $\{T_i\}_1^\infty$, we obtain for any $\epsilon > 0$, there is a neighbourhood $U$ of $\mathbf{0} \in \mathcal{S}(\mathbb{R}^n)$ such that $|\langle T_i, \phi \rangle| < \epsilon$ whenever $\phi \in U$. Thus, $|\langle T_i, \phi_i - \phi \rangle| < \epsilon$ for a large enough $i$. From that we conclude that $\langle T_i, \phi_i \rangle \to \langle T, \phi \rangle$. $\qquad\square$

For any $T \in \mathcal{S}'(\mathbb{R}^n)$ and $\psi \in \mathcal{S}(\mathbb{R}^{n-k})$, let us define $T^\psi \in \mathcal{S}'(\mathbb{R}^k)$ as $\langle T^\psi, \phi \rangle = \langle T, \phi \otimes \psi \rangle$.

Suppose that $\{f_i\}_1^\infty \subseteq \mathcal{S}'(\mathbb{R}^k)$, $\{U_i\}_1^\infty$ are such that $(f_i \otimes \delta^{n-k})_{U_i} \to^* f$. We need to prove that $f \in \mathcal{G}_k'$. Since a set of orthogonal matrices is compact, then one can always find a subsequence $\{U_{n_i}\}$ such that $U_{n_i} \to U$. Since $(f_{n_i} \otimes \delta^{n-k})_{U_{n_i}} \to^* f$ and $\phi(U_{n_i}\mathbf{x}) \to \phi(U\mathbf{x})$ (for any fixed $\phi \in \mathcal{S}(\mathbb{R}^n)$), using lemma 1 we obtain:

$$\langle f_{n_i} \otimes \delta^{n-k}, \phi \rangle = \langle (f_{n_i} \otimes \delta^{n-k})_{U_{n_i}}, \phi(U_{n_i}\mathbf{x}) \rangle \to \langle f, \phi(U\mathbf{x}) \rangle = \langle f_{U^T}, \phi(\mathbf{x}) \rangle$$

Thus, we have:

$$f_{n_i} \otimes \delta^{n-k} \to^* f_{U^T}$$

From the last we see that $f_{n_i} \to^* f_{U^T}^\psi$ where $\psi$ is such that $\psi(\mathbf{0}) = 1$. Therefore, $f_{U^T} = f_{U^T}^\psi \otimes \delta^{n-k}$ and $f = (f_{U^T}^\psi \otimes \delta^{n-k})_U \in \mathcal{G}_k'$. $\qquad\square$

### A.2 Proof of Theorem 2

*Proof.* Let us prove first that if $g = T_f \otimes \delta^{n-k}$, then

$$\mathcal{F}[g] = T_r$$

where $r(\mathbf{x}) = \hat{f}(\mathbf{x}_{1:k}), \mathbf{x} \in \mathbb{R}^n$. For that we have to prove that $\langle \mathcal{F}[g], \phi \rangle = \langle T_r, \phi \rangle$ for any $\phi \in \mathcal{S}(\mathbb{R}^n)$. Indeed,

$$\langle \mathcal{F}[g], \phi \rangle = \langle g, \mathcal{F}[\phi] \rangle = \langle T_f \otimes \delta^{n-k}, \int_{\mathbb{R}^n} \phi(\mathbf{y}) e^{-i\mathbf{x}^T\mathbf{y}} d\mathbf{y} \rangle =$$

$$\langle T_f, \int_{\mathbb{R}^n} \phi(\mathbf{y}) e^{-i\mathbf{x}_{1:k}^T\mathbf{y}_{1:k}} d\mathbf{y} \rangle = \int_{\mathbb{R}^{n+k}} f(\mathbf{x}_{1:k}) \phi(\mathbf{y}) e^{-i\mathbf{x}_{1:k}^T\mathbf{y}_{1:k}} d\mathbf{y} d\mathbf{x}_{1:k} =$$

$$\int_{\mathbb{R}^n} \hat{f}(\mathbf{y}_{1:k}) \phi(\mathbf{y}) d\mathbf{y} = \langle T_r, \phi \rangle$$

Let us calculate the image of $\mathcal{G}_k$ under the Fourier transform. It is easy to see that for any $g \in \mathcal{S}'(\mathbb{R}^n), \phi \in \mathcal{S}(\mathbb{R}^n)$ and orthogonal $U \in \mathbb{R}^{n \times n}$ we have:

$$\langle \mathcal{F}[g_U], \phi(\mathbf{x}) \rangle = \langle g_U, \mathcal{F}[\phi](\mathbf{x}) \rangle = \langle g, \mathcal{F}[\phi](U^T\mathbf{x}) \rangle =$$

$$= \langle g, \mathcal{F}[\phi(U^T\mathbf{x})] \rangle = \langle \mathcal{F}[g], \phi(U^T\mathbf{x}) \rangle = \langle (\mathcal{F}[g])_U, \phi(\mathbf{x}) \rangle$$

Therefore, $\mathcal{F}[g_U] = (\mathcal{F}[g])_U$. Thus, if $g = T_f \otimes \delta^{n-k}$, then

$$(\mathcal{F}[g_U]) = (T_r)_U = T_{r'}$$

where $r'(\mathbf{x}) = r(U\mathbf{x}) = \hat{f}(U_k\mathbf{x})$ and $U_k \in \mathbb{R}^{k \times n}$ is a matrix consisting of first $k$ rows of $U$. Thus, $T_{r'} \in \mathcal{F}_k$.

Let us show that by varying $f \in \mathcal{S}(\mathbb{R}^k)$ and $U$ in the expression $\hat{f}(U_k\mathbf{x})$ we can obtain any function from $\mathcal{F}_k$. For this it is enough to show that $\mathcal{F}_k$ is equivalent to the following set of functions:

$$\mathcal{Q} = \{g(U_k\mathbf{x}) | g \in \mathcal{S}(\mathbb{R}^k), U_k \in \mathbb{R}^{k \times n}, U_k U_k^T = I_k\}$$

The fact $\mathcal{Q} \subseteq \mathcal{F}_k$ is obvious. Let us now prove that $\mathcal{Q} \supseteq \{g(P\mathbf{x}) | g \in \mathcal{S}(\mathbb{R}^k), P \in \mathbb{R}^{k \times n}, \text{rank } P = k\} = \mathcal{F}_k$. Indeed, if $f(\mathbf{x}) = g(P\mathbf{x})$, then $f(\mathbf{x}) = g'(U_k\mathbf{x})$ where $U_k = (PP^T)^{-1/2}P$ and $g'(\mathbf{y}) = g((PP^T)^{1/2}\mathbf{y})$. By construction, $U_k U_k^T = I_k$ and $g' \in \mathcal{S}(\mathbb{R}^k)$. Thus, $\mathcal{Q} = \mathcal{F}_k$.

Therefore, $\mathcal{F}[\mathcal{G}_k] = \mathcal{F}_k$, and from the bijectivity of the Fourier transform we obtain $\mathcal{F}^{-1}[\mathcal{F}_k] = \mathcal{G}_k$. $\qquad\square$

## A.3 PROOF OF THEOREM 3

*Proof of Theorem 3 ($\Rightarrow$).* Let us prove that from $T = (f \otimes \delta^{n-k})_U, f \in \mathcal{S}'(\mathbb{R}^k), U^T U = I_n$ it follows that $\dim \operatorname{span}_{\mathbb{R}}\{x_1 T, x_2 T, \cdots, x_n T\} \leq k$.

It is easy to see that $x_i[f \otimes \delta^{n-k}] = 0$ if $i > k$. If $U = [\mathbf{u}_1, \cdots, \mathbf{u}_n]^T$, then for $i > k$ we have $0 = (x_i[f \otimes \delta^{n-k}])_U = \mathbf{u}_i^T \mathbf{x}(f \otimes \delta^{n-k})_U = \mathbf{u}_i^T \mathbf{x} T$.

Thus, we have $n - k$ orthogonal vectors, $\mathbf{u}_{k+1}, \cdots, \mathbf{u}_n$, such that $[x_1 T, \cdots, x_n T]\mathbf{u}_i = 0$. Using standard linear algebra we obtain there are at most $k'$ distributions $x_{i_1} T, \cdots, x_{i_{k'}} T, k' \leq k$ that form a basis of $\operatorname{span}_{\mathbb{R}}\{x_i T\}_1^n$. $\qquad \square$

To prove the second part of theorem we need the following lemma.

**Lemma 2.** *If $T \in \mathcal{S}'(\mathbb{R}^n)$ is such that $y_i T = 0$ for any $i > k$, then $T \in \mathcal{G}'_k$.*

*Proof of lemma.* Recall from functional analysis, for $f \in \mathcal{S}'(\mathbb{R}^n)$, the tempered distribution $\frac{\partial f}{\partial x_i}$ is defined by the condition $\langle \frac{\partial f}{\partial x_i}, \phi \rangle = -\langle f, \frac{\partial \phi}{\partial x_i} \rangle$. Once the Fourier transform is applied, our lemma's dual version is equivalent to the following formulation: if $\frac{\partial f}{\partial x_i} = 0, i > k$, then $f \in \overline{\mathcal{F}_k}^*$. Let us prove it in this formulation.

A set of infinitely differentiable functions with a compact support is denoted by $C_c^\infty(\mathbb{R})$. Suppose $\phi \in \mathcal{S}(\mathbb{R}^n)$ and $p \in C_c^\infty(\mathbb{R})$ are chosen in such a way that $\int_{-\infty}^\infty p(y_i)dy_i = 1$, $\mathbf{supp}\, p \subseteq [A, B]$. Let us define:

$$r(\mathbf{x}) = \int_{-\infty}^{x_i} \phi(\mathbf{x}_{-i}, y_i)dy_i - \int_{-\infty}^{x_i} p(y_i)dy_i \int_{-\infty}^\infty \phi(\mathbf{x}_{-i}, y_i)dy_i$$

It is easy to see that for any $\alpha \in \mathbb{N}^{n-1}, \alpha' \in \mathbb{N}, \beta \in \mathbb{N}^{n-1}, \beta' \in \mathbb{N}$ we have (at least one derivative over $x_i$ is present):

$$\mathbf{x}_{-i}^\alpha x_i^{\alpha'} \frac{\partial^{\beta, 1+\beta'} r}{\partial \mathbf{x}_{-i}^\beta \partial x_i^{1+\beta'}} = \mathbf{x}_{-i}^\alpha x_i^{\alpha'} \frac{\partial^{\beta, \beta'}[\phi(\mathbf{x}) - p(x_i)\int_{-\infty}^\infty \phi(\mathbf{x}_{-i}, y_i)dy_i]}{\partial \mathbf{x}_{-i}^\beta \partial x_i^{\beta'}} =$$

$$\mathbf{x}_{-i}^\alpha x_i^{\alpha'} \frac{\partial^{\beta, \beta'} \phi(\mathbf{x})}{\partial \mathbf{x}_{-i}^\beta \partial x_i^{\beta'}} - x_i^{\alpha'} \frac{\partial^{\beta'} p(x_i)}{\partial x_i^{\beta'}} \int_{-\infty}^\infty \mathbf{x}_{-i}^\alpha \frac{\partial^\beta \phi(\mathbf{x}_{-i}, y_i)}{\partial \mathbf{x}_{-i}^\beta} dy_i$$

The terms $\mathbf{x}_{-i}^\alpha x_i^{\alpha'} \frac{\partial^{\beta, \beta'} \phi(\mathbf{x})}{\partial \mathbf{x}_{-i}^\beta \partial x_i^{\beta'}}$ and $x_i^{\alpha'} \frac{\partial^{\beta'} p(x_i)}{\partial x_i^{\beta'}}$ are bounded by the definition of $\mathcal{S}(\mathbb{R}^n), C_c^\infty(\mathbb{R})$. The boundedness of $\int_{-\infty}^\infty \mathbf{x}_{-i}^\alpha \frac{\partial^\beta \phi(\mathbf{x}_{-i}, y_i)}{\partial \mathbf{x}_{-i}^\beta} dy_i$ is a consequence of the inequality (which holds because $\phi \in \mathcal{S}(\mathbb{R}^n)$): $|\mathbf{x}_{-i}^\alpha \frac{\partial^\beta \phi(\mathbf{x}_{-i}, y_i)}{\partial \mathbf{x}_{-i}^\beta}| \leq \frac{C}{1+y_i^2}$.

Analogously (not a single derivative over $x_i$ is present):

$$\mathbf{x}_{-i}^\alpha x_i^{\alpha'} \frac{\partial^\beta r}{\partial \mathbf{x}_{-i}^\beta} = x_i^{\alpha'} \int_{-\infty}^{x_i} \mathbf{x}_{-i}^\alpha \frac{\partial^\beta \phi(\mathbf{x}_{-i}, y_i)}{\partial \mathbf{x}_{-i}^\beta} dy_i - x_i^{\alpha'} \int_{-\infty}^{x_i} p(y_i)dy_i \int_{-\infty}^\infty \mathbf{x}_{-i}^\alpha \frac{\partial^\beta \phi(\mathbf{x}_{-i}, y_i)}{\partial \mathbf{x}_{-i}^\beta} dy_i =$$

$$= x_i^{\alpha'}(1 - \int_{-\infty}^{x_i} p(y_i)dy_i) \int_{-\infty}^{x_i} \mathbf{x}_{-i}^\alpha \frac{\partial^\beta \phi(\mathbf{x}_{-i}, y_i)}{\partial \mathbf{x}_{-i}^\beta} dy_i - x_i^{\alpha'} \int_{-\infty}^{x_i} p(y_i)dy_i \int_{x_i}^\infty \mathbf{x}_{-i}^\alpha \frac{\partial^\beta \phi(\mathbf{x}_{-i}, y_i)}{\partial \mathbf{x}_{-i}^\beta} dy_i$$

The second term is 0 when $x_i \leq A$. It is also bounded when $x_i > A$ because $|\mathbf{x}_{-i}^\alpha \frac{\partial^\beta \phi(\mathbf{x}_{-i}, y_i)}{\partial \mathbf{x}_{-i}^\beta}| \leq \frac{C'}{(1+y_i^2)^{\alpha'+1}}$ and:

$$\left| x_i^{\alpha'} \int_{x_i}^\infty \mathbf{x}_{-i}^\alpha \frac{\partial^\beta \phi(\mathbf{x}_{-i}, y_i)}{\partial \mathbf{x}_{-i}^\beta} dy_i \right| \leq |x_i|^{\alpha'} \int_{x_i}^\infty \frac{C'}{(1+y_i^2)^{\alpha'+1}} dy_i$$

The latter is bounded because $\lim_{x_i \to +\infty} |x_i|^{\alpha'} \int_{x_i}^\infty \frac{C'}{(1+y_i^2)^{\alpha'+1}} dy_i = 0$.

The first term is 0 when $x_i \geq B$ and it is bounded for $x_i < B$:

$$\left| x_i^{\alpha'} \int_{-\infty}^{x_i} \mathbf{x}_{-i}^{\alpha} \frac{\partial^{\beta} \phi(\mathbf{x}_{-i}, y_i)}{\partial \mathbf{x}_{-i}^{\beta}} dy_i \right| \leq |x_i|^{\alpha'} \int_{-\infty}^{x_i} \frac{C'}{(1 + y_i^2)^{\alpha'+1}} dy_i$$

The latter is also bounded, since $\lim_{x_i \to -\infty} |x_i|^{\alpha'} \int_{-\infty}^{x_i} \frac{C'}{(1+y_i^2)^{\alpha'+1}} dy_i = 0$.

Thus, $\mathbf{x}^{\alpha} \frac{\partial^{\beta} r(\mathbf{x})}{\partial \mathbf{x}^{\beta}}$ is bounded and $r \in \mathcal{S}(\mathbb{R}^n)$. Therefore $\frac{\partial f}{\partial x_i} = 0$ implies:

$$\langle f, \frac{\partial r}{\partial x_i} \rangle = 0 \Rightarrow f[\phi] = f[p(x_i) \int_{-\infty}^{\infty} \phi(\mathbf{x}_{-i}, y_i) dy_i]$$

Since this sequence of arguments works for any $i > k$, we can apply them sequentially to initial $\phi \in \mathcal{S}(\mathbb{R}^n)$ w.r.t. $x_{k+1}, ..., x_n$ and obtain for any $p_{k+1}, ..., p_n \in C_c(\mathbb{R})$ such that $\int_{-\infty}^{\infty} p_i(y_i) dy_i = 1$:

$$f[\phi] = f[p_{k+1}(x_{k+1}) \cdots p_n(x_n) \int_{\mathbb{R}^{n-k}} \phi(\mathbf{x}_{1:k}, \mathbf{x}_{k+1:n}) d\mathbf{x}_{k+1:n}]$$

Moreover, since $C_c^{\infty}(\mathbb{R})$ is dense in $\mathcal{S}(\mathbb{R})$, we can assume that $p_{k+1}, ..., p_n \in \mathcal{S}(\mathbb{R})$. For the inverse Fourier transform $T = \mathcal{F}^{-1}[f]$ the latter condition becomes equivalent to:

$$\langle T, \phi \rangle = \langle T, p'_{k+1}(x_{k+1}) \cdots p'_n(x_n) \phi(\mathbf{x}_{1:k}, \mathbf{0}_{k+1:n}) \rangle$$

for any $p'_{k+1}, ..., p'_n \in \mathcal{S}(\mathbb{R})$ such that $p'_i(0) = 1$. Let us define $p'_i(x_i) = e^{-x_i^2}$. It is easy to check that $T = g \otimes \delta^{n-k}$ where $g \in \mathcal{S}'(\mathbb{R}^k)$, $\langle g, \psi \rangle = \langle T, e^{-|\mathbf{x}_{k+1:n}|^2} \psi(\mathbf{x}_{1:k}) \rangle$ for $\psi \in \mathcal{S}(\mathbb{R}^k)$. I.e. $T \in \mathcal{G}'_k$ and lemma is proved. $\square$

*Proof of Theorem 3 ($\Leftarrow$).* If $\dim \text{span}_{\mathbb{R}} \{ x_1 T, x_2 T, \cdots, x_n T \} \leq k$, then

$$\dim \{ \mathbf{v} \in \mathbb{R}^n | [x_1 T, \cdots, x_n T] \mathbf{v} = 0 \} \geq n - k$$

Thus, there exist at least $n - k$ orthonormal vectors $\mathbf{v}_{k+1}, \cdots, \mathbf{v}_n$, such that $[x_1 T, \cdots, x_n T] \mathbf{v}_i = 0$. Therefore, $[x_1 T, \cdots, x_n T] \mathbf{v}_i = (\mathbf{v}_i^T \mathbf{x}) T = 0$.

Let us complete $\mathbf{v}_{k+1}, \cdots, \mathbf{v}_n$ to form an orthonormal basis of $\mathbb{R}^n$: $\mathbf{v}_1, \cdots, \mathbf{v}_n$. Let us define a matrix $V = [\mathbf{v}_1, \cdots, \mathbf{v}_n]$. It is easy to see that:

$$\left( (\mathbf{v}_i^T \mathbf{x}) T \right)_V = (\mathbf{v}_i^T V \mathbf{x}) T_V = x_i T_V$$

Since for $i > k$ we have $(\mathbf{v}_i^T \mathbf{x}) T = 0$, then $x_i T_V = 0$. Using lemma 2 we obtain $T_V \in \mathcal{G}'_k$. Therefore, $(T_V)_{V^T} = T \in \mathcal{G}'_k$. Theorem proved. $\square$

## B  STRUCTURE OF WD

Recall that $(\mathbb{R}^n, || \cdot ||)$ is a Banach space. Now, let us consider an optimization problem: for a given $X \in \mathbb{R}^{n \times N}$ solve

$$||X - L|| \to \min_{\text{rank}(L) \leq k} \tag{13}$$

where $|| \cdot ||$ is extended to $\mathbb{R}^{n \times N}$ by $||[\mathbf{s}_1, \cdots, \mathbf{s}_N]|| \overset{def}{=} \sum_i ||\mathbf{s}_i||$.

The following simple theorem shows that the two tasks are connected, so that the solution of one directly leads to the solution of another.

**Theorem 6.** *Given data points $\{\mathbf{x}_1, \cdots, \mathbf{x}_N\}$, let $X = [\mathbf{x}_1, \cdots, \mathbf{x}_N] \in \mathbb{R}^{n \times N}$. Then,*

$$\min_{\nu \in \mathcal{P}_k} W(\mu_{\text{data}}, \nu) = \frac{1}{N} \min_{Y \in \mathbb{R}^{n \times N}, \text{rank}(Y) \leq k} ||X - Y||$$

*Moreover, $\min_{\nu \in \mathcal{P}_k} W(\mu_{\text{data}}, \nu)$ is attained on $\nu^*$, where $\nu^*$ is a uniform distribution over $\{\mathbf{y}_i\}_{i=1}^N$ and $[\mathbf{y}_1, \cdots, \mathbf{y}_N] \in \arg\min_{Y \in \mathbb{R}^{n \times N}, \text{rank}(Y) \leq k} ||X - Y||$.*

*Proof.* Let us prove first that $\inf_{\mu \in \mathcal{P}_k} W(\mu_{\text{data}}, \mu) \leq \frac{1}{N}||X - Y^*||$ where

$$Y^* = [\mathbf{y}_1, \cdots, \mathbf{y}_N] \in \arg \min_{Y \in \mathbb{R}^{n \times N}, \text{rank}(Y) \leq k} ||X - Y||$$

Let $\pi$ be a uniform distribution over $\{(\mathbf{x}_i, \mathbf{y}_i)\}_{i=1}^N$ and $\mu^*$ be a uniform distribution over $\{\mathbf{y}_i\}_{i=1}^N$. Since $\pi \in \Pi(\mu_{\text{data}}, \mu^*)$, we obtain $W(\mu_{\text{data}}, \mu^*) \leq \frac{1}{N} \sum_{i=1}^N ||\mathbf{x}_i - \mathbf{y}_i|| = \frac{1}{N}||X - Y^*||$. The support of $\mu^*$ is $k$-dimensional, because $\text{rank}(Y^*) \leq k$. Thus, we have $\mu^* \in \mathcal{P}_k$ and $\inf_{\mu \in \mathcal{P}_k} W(\mu_{\text{data}}, \mu) \leq W(\mu_{\text{data}}, \mu^*) \leq \frac{1}{N}||X - Y^*||$. Now, if we prove the inverse inequality, i.e. $\inf_{\mu \in \mathcal{P}_k} W(\mu_{\text{data}}, \mu) \geq \frac{1}{N}||X - Y^*||$, this will imply that $\inf_{\mu \in \mathcal{P}_k} W(\mu_{\text{data}}, \mu) = \frac{1}{N}||X - Y^*||$ and therefore, $\inf_{\mu \in \mathcal{P}_k} W(\mu_{\text{data}}, \mu) = W(\mu_{\text{data}}, \mu^*)$. This will in the end give us $\mu^* \in \arg\inf_{\mu \in \mathcal{P}_k} W(\mu_{\text{data}}, \mu)$.

Let $\{\mu_t\}_1^\infty$ be such that $\mu_t \in \mathcal{P}_k$ and $W(\mu_{\text{data}}, \mu_t) - \inf_{\mu \in \mathcal{P}_k} W(\mu_{\text{data}}, \mu) \to 0$. Let $L_t$ denote a $k$-dimensional support of $\mu_t$ and $P_t$ is a projection operator onto $L_t$.

Let $\mu_t^*$ be a uniform distribution over $\{P_t \mathbf{x}_1, \cdots, P_t \mathbf{x}_N\}$, i.e. $\mu_t^*(A) = \frac{1}{N} \sum_{i=1}^N [P_t \mathbf{x}_i \in A]$. It is easy to see that $W(\mu_t^*, \mu_{\text{data}}) \leq W(\mu_t, \mu_{\text{data}})$, because $\mu_t^*$ and $\mu_t$ share the same $k$-dimensional support $L_t$, but the "transportation of a mass" concentrated in point $\mathbf{x}_i$ of the empirical distribution $\mu_{\text{emp}}$ can be most optimally done by just moving it to $P_t \mathbf{x}_i$ (i.e. to the closest point on $L_t$). Thus, we have $\inf_{\mu \in \mathcal{P}_k} W(\mu_{\text{data}}, \mu) \leq W(\mu_{\text{data}}, \mu_t^*) \leq W(\mu_{\text{data}}, \mu_t)$, and therefore, $W(\mu_{\text{data}}, \mu_t^*) - \inf_{\mu \in \mathcal{P}_k} W(\mu_{\text{data}}, \mu) \to 0$.

Since a set of projection operators is compact, one can always extract a subsequence $\{P_{t_s}\}_{s=1}^\infty$, such that $P_{t_s} \to P$. It is easy to see that $\mu_{t_s}^* \to \mu^{**}$ (i.e. $W(\mu_{t_s}^*, \mu^{**}) \to 0$) where $\mu^{**}$ is a uniform distribution over $\{P\mathbf{x}_1, \cdots, P\mathbf{x}_N\}$. For that distribution we have $W(\mu_{\text{data}}, \mu^{**}) = \lim_{s \to \infty} W(\mu_{\text{data}}, \mu_{t_s}^*) = \inf_{\mu \in \mathcal{P}_k} W(\mu_{\text{data}}, \mu)$. Thus, the infimum is attained on $\mu^{**}$.

It is easy to see that $W(\mu_{\text{data}}, \mu^{**}) = W(\mu^{**}, \mu_{\text{data}}) = \frac{1}{N}||X - PX||$. Since $\text{rank}(PX) \leq k$ we obtain $W(\mu_{\text{data}}, \mu^{**}) \geq \frac{1}{N} \min_{Y \in \mathbb{R}^{n \times N}, \text{rank}(Y) \leq k} ||X - Y||$. This completes the proof. $\square$

Note that the case of $L_1$ norm $||\mathbf{x}|| = \sum_i |x_i|$ in the task 13 corresponds to the well-studied *robust PCA* problem Candès et al. (2011). If, instead of the $L_1$-norm, we use the $L_2$-norm, this leads to another task:

$$||X - L||_{1,2} \to \min_{\text{rank}(L) \leq k} \tag{14}$$

where $||S||_{1,2} = \sum_j \sqrt{\sum_i s_{ij}^2}$, which known as *the outlier pursuit* problem Xu et al. (2010).

## C  PROPER KERNELS AND PROOF OF THEOREM 4

### C.1  PROPER KERNELS

In the main part of the paper we assume $M$ to be a gaussian kernel, though the theory can be applied to a more general case of the so called *proper kernels*.

Recall that for any operator $O$ between spaces $\mathcal{H}_1$ and $\mathcal{H}_2$ we denote its range as $\mathcal{R}[O] = \{O(x) | x \in \mathcal{H}_1\}$. For $\Omega \subseteq \mathcal{S}(\mathbb{R}^n)$, $\overline{\Omega}$ denotes the sequential closure of $\Omega$ with respect to natural topology of $\mathcal{S}(\mathbb{R}^n)$. A set of continuous functions in $\mathbb{R}^n$ is denoted by $C(\mathbb{R}^n)$. A set of infinitely differentiable functions with compact support in $\mathbb{R}^n$ is denoted as $C_c^\infty(\mathbb{R}^n)$

**Definition 2.** *The function* $M(\mathbf{x}, \mathbf{y}) : \mathbb{R}^n \times \mathbb{R}^n \to \mathbb{C}$ *is called the* proper kernel *if and only if*

- $O(M)[f] = \int_{\mathbb{R}^n} M(\mathbf{x}, \mathbf{y}) f(\mathbf{y}) d\mathbf{y}$ *is a linear operator from* $L_2(\mathbb{R}^n)$ *to* $L_2(\mathbb{R}^n)$,

- $M(\mathbf{y}, \mathbf{x}) = M(\mathbf{x}, \mathbf{y})^*$,

- $\langle f, O(M)[f] \rangle_{L_2(\mathbb{R}^n)} > 0, \forall f \in L_2(\mathbb{R}^n), f \neq \mathbf{0}$.

- $\max_{\mathbf{x}, \mathbf{y}} |M(\mathbf{x}, \mathbf{y})| < \infty$,

- $\overline{\mathcal{R}[O(M)] \cap \mathcal{S}(\mathbb{R}^n)} = \mathcal{S}(\mathbb{R}^n)$.

The gaussian kernel $M(\mathbf{x}, \mathbf{y}) = G_\sigma^n(\mathbf{x} - \mathbf{y})$, which is of special interest from an application-oriented perspective, is captured by the following lemma:

**Lemma 3.** *If $\zeta, \hat{\zeta} \in C(\mathbb{R}^n)$ are bounded, $\forall \mathbf{x} \; \hat{\zeta}(\mathbf{x}) > 0$, then $M(\mathbf{x}, \mathbf{y}) = \zeta(\mathbf{x} - \mathbf{y})$ is a proper kernel.*

*Proof.* Verification of the first four conditions is easy, so we only check the fifth condition. Let us denote linear operators $C_\zeta[f] = \zeta * f$ and $O_g[f](\mathbf{x}) = g(\mathbf{x})f(\mathbf{x})$. Then we have $\mathcal{F}[C_\zeta[L_2(\mathbb{R}^n)]] = O_{\hat{\zeta}}[L_2(\mathbb{R}^n)] \supseteq C_c^\infty(\mathbb{R}^n)$. Therefore, $\mathcal{R}[O(M)] = C_\zeta[L_2(\mathbb{R}^n)] \supseteq \mathcal{F}^{-1}[C_c^\infty(\mathbb{R}^n)]$. Since $C_c^\infty(\mathbb{R}^n)$ is dense in $\mathcal{S}(\mathbb{R}^n)$, then $\mathcal{F}^{-1}[C_c^\infty(\mathbb{R}^n)]$ also has this property. I.e. $\overline{\mathcal{R}[O(M)] \cap \mathcal{S}(\mathbb{R}^n)} = \mathcal{S}(\mathbb{R}^n)$. $\square$

Besides the gaussian kernel the lemma also captures a case of the Abel Kernel $\zeta(\mathbf{x}) = e^{-|\mathbf{x}|}$. It is well-known that the Fourier tranform of the Abel Kernel is the Poisson kernel: $\hat{\zeta}(\mathbf{x}) = \dfrac{c_n}{(1+|\mathbf{x}|^2)^{\frac{n+1}{2}}}$ (which is also proper).

## C.2 Proof of Theorem 4

We will prove a more general statement:

**Theorem 7.** *Let $M(\mathbf{x}, \mathbf{y})$ be a proper kernel and, additionally, a Lipschitz function. If $f \in \mathcal{G}_k$, then $\langle x_i f | M | x_j f \rangle$ is defined and $\operatorname{rank} M_f \leq k$.*

*Proof.* Let us first show that $\langle f | M | g \rangle$ is defined for all $f, g \in \mathcal{G}_k$. Note that for any $f = (T_a \otimes \delta^{n-k})_U \in \mathcal{G}_k$ we have

$$T_{f_\epsilon} = (T_a \otimes \delta^{n-k})_U * G_\epsilon^n = ((T_a * G_\epsilon^k) \otimes T_{G_\epsilon^{n-k}})_U$$

Let us denote $a_\epsilon = a * G_\epsilon^k$ and $b_\epsilon = b * G_\epsilon^k$. It is easy to see that

$$f_\epsilon = (a_\epsilon(\mathbf{x}_{1:k})G_\epsilon^{n-k}(\mathbf{x}_{k+1:n}))_U \in \mathcal{S}(\mathbb{R}^n)$$

From a well-known property of the Weierstrass transform we have:

$$||f_\epsilon||_{L_1} = ||a_\epsilon||_{L_1} \cdot ||G_\epsilon^{n-k}||_{L_1} \leq ||a||_{L_1}$$

From this we obtain for any $f = (T_a \otimes \delta^{n-k})_U, g = (T_b \otimes \delta^{n-k})_V \in \mathcal{G}_k$:

$$|\langle f_\epsilon | M | g_\epsilon \rangle| \leq \max_{\mathbf{x}, \mathbf{y}} |M(\mathbf{x}, \mathbf{y})| \; ||f_\epsilon||_{L_1} ||g_\epsilon||_{L_1} \leq \max_{\mathbf{x}, \mathbf{y}} |M(\mathbf{x}, \mathbf{y})| \; ||a||_{L_1} ||b||_{L_1} < \infty$$

Thus, $\langle f_\epsilon | M | g_\epsilon \rangle$ is defined and:

$$\langle f_\epsilon | M | g_\epsilon \rangle = \int\limits_{\mathbb{R}^n \times \mathbb{R}^n} a_\epsilon^*(\mathbf{x}_{1:k})G_\epsilon^{n-k}(\mathbf{x}_{k+1:n})M(U^T\mathbf{x}, V^T\mathbf{y})b_\epsilon(\mathbf{y}_{1:k})G_\epsilon^{n-k}(\mathbf{y}_{k+1:n})d\mathbf{x}d\mathbf{y} =$$

$$= \int\limits_{\mathbb{R}^k \times \mathbb{R}^k} a_\epsilon^*(\mathbf{x}_{1:k})M_\epsilon(\mathbf{x}_{1:k}, \mathbf{y}_{1:k})b_\epsilon(\mathbf{y}_{1:k})d\mathbf{x}_{1:k}d\mathbf{y}_{1:k}$$

where

$$M_\epsilon(\mathbf{x}_{1:k}, \mathbf{y}_{1:k}) = \int\limits_{\mathbb{R}^{n-k} \times \mathbb{R}^{n-k}} G_\epsilon^{n-k}(\mathbf{x}_{k+1:n})M(U^T\mathbf{x}, V^T\mathbf{y})G_\epsilon^{n-k}(\mathbf{y}_{k+1:n})d\mathbf{x}_{k+1:n}d\mathbf{y}_{k+1:n}$$

Let $U_k, V_k \in \mathbb{R}^{n \times n}$ be matrices that comprise the first $k$ rows of $U, V$ correspondingly and $n - k$ zero rows below. Also, let $L$ denote Lipschitz constant for $M$ such that $|M(\mathbf{x}, \mathbf{y}) - M(\mathbf{x}', \mathbf{y}')| \leq L(|\mathbf{x} - \mathbf{x}'| + |\mathbf{y} - \mathbf{y}'|)$. For the function $M_\epsilon(\mathbf{x}_{1:k}, \mathbf{y}_{1:k})$ we have:

$$|M_\epsilon(\mathbf{x}_{1:k}, \mathbf{y}_{1:k}) - M(U_k^T\mathbf{x}, V_k^T\mathbf{y})| =$$

$$|\int\limits_{\mathbb{R}^{2n-2k}} G_\epsilon^{n-k}(\mathbf{x}_{k+1:n}) \left( M(U^T\mathbf{x}, V^T\mathbf{y}) - M(U_k^T\mathbf{x}, V_k^T\mathbf{y}) \right) G_\epsilon^{n-k}(\mathbf{y}_{k+1:n})d\mathbf{x}_{k+1:n}d\mathbf{y}_{k+1:n}|$$

$$\leq L|\int_{\mathbb{R}^{2n-2k}} G_\epsilon^{n-k}(\mathbf{x}_{k+1:n})\left(|(U-U_k)^T\mathbf{x}|+|(V-V_k)^T\mathbf{y}|\right)G_\epsilon^{n-k}(\mathbf{y}_{k+1:n})d\mathbf{x}_{k+1:n}d\mathbf{y}_{k+1:n}|$$

$$=L|\int_{\mathbb{R}^{2n-2k}} G_\epsilon^{n-k}(\mathbf{x}_{k+1:n})\left(|\mathbf{x}_{k+1:n}|+|\mathbf{y}_{k+1:n}|\right)G_\epsilon^{n-k}(\mathbf{y}_{k+1:n})d\mathbf{x}_{k+1:n}d\mathbf{y}_{k+1:n}|=$$

$$=2L\int_{\mathbb{R}^{n-k}}|\mathbf{x}_{k+1:n}|G_\epsilon^{n-k}(\mathbf{x}_{k+1:n})d\mathbf{x}_{k+1:n}=2L\epsilon\int_{\mathbb{R}^{n-k}}|\mathbf{x}_{k+1:n}|G_1^{n-k}(\mathbf{x}_{k+1:n})d\mathbf{x}_{k+1:n}$$

Thus, there exists bounded $\tilde{M}(\mathbf{x}_{1:k},\mathbf{y}_{1:k})=M(U_k^T\mathbf{x},V_k^T\mathbf{y})$ such that:

$$M_\epsilon(\mathbf{x}_{1:k},\mathbf{y}_{1:k})\overset{\epsilon\to 0}{\to}\tilde{M}(\mathbf{x}_{1:k},\mathbf{y}_{1:k})\text{ in }L_\infty(\mathbb{R}^{2k})$$

Further we assume that $\epsilon>0$ is small enough, so that $M_\epsilon(\mathbf{x}_{1:k},\mathbf{y}_{1:k})\leq C=2\max|M|$. Now we have:

$$|\langle f_\epsilon|M|g_\epsilon\rangle-\int_{\mathbb{R}^k\times\mathbb{R}^k}a^*(\mathbf{x}_{1:k})\tilde{M}(\mathbf{x}_{1:k},\mathbf{y}_{1:k})b(\mathbf{y}_{1:k})d\mathbf{x}_{1:k}d\mathbf{y}_{1:k}|=$$

$$|\int_{\mathbb{R}^k\times\mathbb{R}^k}(a_\epsilon^*(\mathbf{x}_{1:k})M_\epsilon(\mathbf{x}_{1:k},\mathbf{y}_{1:k})b_\epsilon(\mathbf{y}_{1:k})-a^*(\mathbf{x}_{1:k})\tilde{M}(\mathbf{x}_{1:k},\mathbf{y}_{1:k})b(\mathbf{y}_{1:k}))d\mathbf{x}_{1:k}d\mathbf{y}_{1:k}|=$$

$$|\int_{\mathbb{R}^k\times\mathbb{R}^k}M_\epsilon(\mathbf{x}_{1:k},\mathbf{y}_{1:k})a_\epsilon^*(\mathbf{x}_{1:k})(b_\epsilon(\mathbf{y}_{1:k})-b(\mathbf{y}_{1:k}))d\mathbf{x}_{1:k}d\mathbf{y}_{1:k}+$$

$$\int_{\mathbb{R}^k\times\mathbb{R}^k}M_\epsilon(\mathbf{x}_{1:k},\mathbf{y}_{1:k})b(\mathbf{y}_{1:k})(a_\epsilon^*(\mathbf{x}_{1:k})-a^*(\mathbf{x}_{1:k}))d\mathbf{x}_{1:k}d\mathbf{y}_{1:k}+$$

$$\int_{\mathbb{R}^k\times\mathbb{R}^k}a^*(\mathbf{x}_{1:k})b(\mathbf{y}_{1:k})(M_\epsilon(\mathbf{x}_{1:k},\mathbf{y}_{1:k})-\tilde{M}(\mathbf{x}_{1:k},\mathbf{y}_{1:k}))d\mathbf{x}_{1:k}d\mathbf{y}_{1:k}|\leq$$

$$C||a_\epsilon||_{L_1}||b_\epsilon-b||_{L_1}+C||b||_{L_1}||a_\epsilon-a||_{L_1}+||a^*(\mathbf{x}_{1:k})b(\mathbf{y}_{1:k})||_{L_1}||M_\epsilon-\tilde{M}||_{L_\infty}$$

It is well-known (e.g. see Theorem 2.25) that $||a_\epsilon-a||_{L_p}$, $||b_\epsilon-b||_{L_p}\to 0$, $||a_\epsilon||_{L_1}\leq||a||_{L_1}$ and $||M_\epsilon-\tilde{M}||_{L_\infty}\to 0$. Thus, $\lim_{\epsilon\to 0}\langle f_\epsilon|M|g_\epsilon\rangle$ exists and $\langle f|M|g\rangle$ is defined.

Let us now prove that $\text{rank}\,M_f\leq k$. Since $f\in\mathcal{G}_k$, then $f=(T_g\otimes\delta^{n-k})_U$ where $U$ is an orthogonal matrix and $U=[\mathbf{w}_1,\cdots,\mathbf{w}_n]$. It is easy to see that:

$$\langle x_i f|M|x_j f\rangle=\langle(x_i f)_{U^T}|M(U^T\mathbf{x},U^T\mathbf{y})|(x_j f)_{U^T}\rangle=$$

$$\langle\mathbf{w}_i^T\mathbf{x}T_g\otimes\delta^{n-k}|M(U^T\mathbf{x},U^T\mathbf{y})|\mathbf{w}_j^T\mathbf{x}T_g\otimes\delta^{n-k}\rangle$$

Let us now denote $V=[\mathbf{u}_1,\cdots,\mathbf{u}_n]\in\mathbb{R}^{k\times n}$ a submatrix of $U$ in which only first $k$ rows of $U$ are present. Then, the latter integral is equal to:

$$\iint_{\mathbb{R}^k\times\mathbb{R}^k}\mathbf{u}_i^T\mathbf{x}_{1:k}\mathbf{y}_{1:k}^T\mathbf{u}_j g(\mathbf{x}_{1:k})^*M(V^T\mathbf{x}_{1:k},V^T\mathbf{y}_{1:k})g(\mathbf{y}_{1:k})d\mathbf{x}_{1:k}d\mathbf{y}_{1:k}=\mathbf{u}_i^T B\mathbf{u}_j$$

where

$$B=[\langle x_i g|M'|x_j g\rangle]_{1\leq i,j\leq k},\quad M'(\mathbf{x}_{1:k},\mathbf{y}_{1:k})=M(V^T\mathbf{x}_{1:k},V^T\mathbf{y}_{1:k})$$

is the Gram matrix of the collection $\{x_i g(\mathbf{x}_{1:k})\}_{i=1}^k\subseteq\mathcal{S}(\mathbb{R}^k)$.

Obviously, $\text{rank}\,M_f=\text{rank}\left[\text{Re}\,\mathbf{u}_i^T B\mathbf{u}_j\right]_{1\leq i,j\leq n}=\text{rank}\,V^T(\text{Re}\,B)V\leq\text{rank}\,V=k$. $\qquad\square$

# D  GENERAL THEORY OF THE REDUCTION FOR SECTION 5

For a sequence $\{f_s\}_{s=1}^\infty\subseteq\mathcal{S}'(\mathbb{R}^n)$, $\underset{s\to\infty}{\text{Lim}}\,f_s$ denotes a set of points $f\in\mathcal{S}'(\mathbb{R}^n)$, such that there exists a growing sequence $\{s_i\}\subseteq\mathbb{N}$ and $\lim_{i\to\infty}f_{s_i}=f$.

### D.1 Regular solutions and reduction theorems

For $I : \mathcal{G}'_k \cup \mathcal{S}(\mathbb{R}^n) \to \mathbb{R}^+$, it is natural to reduce the optimization task 10 to an optimization task over ordinary functions with a penalty term 11. To have an equivalence between 10 and 11 we need to assume that $I$'s behaviour when approaching $f \in \mathcal{G}'_k$ from a set $\mathcal{S}(\mathbb{R}^n)$ is continuous, i.e. for any sequence $\{f_i\} \subseteq \mathcal{S}(\mathbb{R}^n)$ such that $T_{f_i} \to^* f \in \mathcal{G}'_k$, we have $\lim_{i\to\infty} I(T_{f_i}) = I(f)$.

Let us introduce the notion of a regular solution both for 10 and 11. Let

$$\mathcal{B}_k = \bigcup_{C>0} \overline{\{f \in \mathcal{G}_k | \operatorname{Tr}(M_f) \leq C\}}^*$$

**Definition 3.** *Any $f \in \operatorname{Arg} \min\limits_{f \in \mathcal{G}'_k} I(f) \bigcap \mathcal{B}_k$ is called a regular solution of 10.*

In other words, $\mathcal{B}_k$ formalizes a set of distributions from $\mathcal{G}'_k$, that can be approached through sequences $\{f_i\} \subseteq \mathcal{G}_k$, for which $\operatorname{Tr}(M_{f_i})$ does not blow up. Obviously, $\mathcal{G}_k \subseteq \mathcal{B}_k \subseteq \mathcal{G}'_k$. In applications, regular solutions include all $\operatorname{Arg} \min\limits_{f \in \mathcal{G}'_k} I(f)$ if we choose the kernel $M$ correctly. This regularity is important for a reduction to the penalty form 11, because when approaching a non-regular solution we are unable to guarantee a bounded behaviour of $M_f$ (and of $R(f)$).

**Definition 4.** *A sequence $\{f_i\}_1^\infty \subseteq \mathcal{S}(\mathbb{R}^n)$ is said to solve 11 if:*

$$I(f_i) + \lambda_i R(f_i) \leq \inf_{f \in \mathcal{S}(\mathbb{R}^n)} I(f) + \lambda_i R(f) + \epsilon_i \tag{15}$$

*where $\epsilon_i \to +0$ and $\lambda_i \to +\infty, i \to +\infty$. If, additionally, $\operatorname{Tr}(M_{f_i})$ is bounded, then $\{f_i\}_1^\infty$ is said to solve 11 regularly.*

Let us define

$$\operatorname{rsol}(I(f), R(f)) = \bigcup_{\{f_i\}_1^\infty \text{ r. solves (11)}} \operatorname{Lim}_{i\to\infty} T_{f_i}$$

**Theorem 8.** *If $M$ is a proper kernel, then $\operatorname{rsol}(I(f), R(f)) \subseteq \operatorname{Arg} \min\limits_{f \in \mathcal{G}'_k} I(f)$.*

**Theorem 9.** *If $M$ is a proper kernel and $\operatorname{rsol}(I(f), R(f)) \neq \emptyset$, then $\operatorname{Arg} \min\limits_{f \in \mathcal{G}'_k} I(f) \bigcap \mathcal{B}_k \subseteq \operatorname{rsol}(I(f), R(f))$.*

**Theorem 10** (Reduction theorem)**.** *If $M$ is a proper kernel, $\operatorname{Arg} \min\limits_{f \in \mathcal{G}'_k} I(f) \subseteq \mathcal{B}_k$ and $\operatorname{rsol}(I(f), R(f)) \neq \emptyset$, then $\operatorname{rsol}(I(f), R(f)) = \operatorname{Arg} \min\limits_{f \in \mathcal{G}'_k} I(f)$.*

Suppose that we now solve a sequence of problems 11 and find $\{f_s\}_1^\infty$. According to Theorems 8 and 9, the following are potential scenarios:

(1) $\operatorname{Tr}(M_{f_s})$ blows up and the convergence is not guaranteed. This situation can be avoided by controlling $\operatorname{Tr}(M_f)$ in an optimization process. In practice, when $f$ has a parameterized form, this can be done by bounding parameters.

If $\operatorname{Tr}(M_{f_s})$ does not blow up, we still have two subcases:

(2.1) $\operatorname{Lim}_{s\to\infty} T_{f_s} \neq \emptyset$. This implies a positive outcome to approach 11 to the optimization problem, Problem 10.

(2.2) $\operatorname{Lim}_{s\to\infty} T_{f_s} = \emptyset$. This exotic situation can happen only if a sequence $T_{f_s}$ leaves any sequentially compact subset of $\mathcal{S}'(\mathbb{R}^n)$. Bounding parameters also tackles this case.

### D.2 Proofs of Theorem 8 and 9

For any $f = (T_l \otimes \delta^{n-k})_U \in \mathcal{G}_k$ and $\sigma > 0$, let us define $f_\sigma$ as:

$$T_{f_\sigma} = (T_l \otimes \delta^{n-k})_U * G_\sigma^n = (T_{l_\sigma} \otimes T_{G_\sigma^{n-k}})_U$$

$$f_\sigma = (l_\sigma(\mathbf{x}_{1:k}) G_\sigma^{n-k}(\mathbf{x}_{k+1:n}))_U$$

$$l_\sigma = l * G_\sigma^k$$

We have $T_{f_\sigma} \to^* (T_l \otimes \delta^{n-k})_U$ as $\sigma \to +0$.

**Lemma 4.** *For any $f \in \mathcal{G}_k$, $\lim_{\sigma \to +0} \langle x_i f_\sigma | M | x_j f_\sigma \rangle = 0$, for any $(i,j) \notin \{1,...,k\}^2$, and $\sup_{\sigma \in [0,1]} \langle x_i f_\sigma | M | x_j f_\sigma \rangle < \infty$, for any $(i,j) \in \{1,...,k\}^2$.*

*Proof.* W.l.o.g. we can assume that $f = T_l \otimes \delta^{n-k}, l \in \mathcal{S}(\mathbb{R}^k)$. If $i > k, j \leq k$ we have:

$$\langle x_i f_\sigma | M | x_j f_\sigma \rangle = \frac{1}{(2\pi\sigma^2)^{n-k}} \iint_{\mathbb{R}^n \times \mathbb{R}^n} x_i y_j e^{-\frac{|\mathbf{x}_{k+1:n}|^2}{2\sigma^2}} l_\sigma(\mathbf{x}_{1:k}) M(\mathbf{x}, \mathbf{y}) e^{-\frac{|\mathbf{y}_{k+1:n}|^2}{2\sigma^2}} l_\sigma(\mathbf{y}_{1:k}) d\mathbf{x} d\mathbf{y} =$$

$$\int_{\mathbb{R}^n} \frac{1}{\sqrt{2\pi\sigma^2}^{n-k}} x_i e^{-\frac{|\mathbf{x}_{k+1:n}|^2}{2\sigma^2}} l_\sigma(\mathbf{x}_{1:k}) P(\mathbf{x}) d\mathbf{x}$$

where $P(\mathbf{x}) = \int_{\mathbb{R}^n} \frac{1}{\sqrt{2\pi\sigma^2}^{n-k}} y_j M(\mathbf{x}, \mathbf{y}) e^{-\frac{|\mathbf{y}_{k+1:n}|^2}{2\sigma^2}} l_\sigma(\mathbf{y}_{1:k}) d\mathbf{y}$. Using the Hólder inequality we obtain:

$$|\langle x_i f_\sigma | M | x_j f_\sigma \rangle| \leq ||\frac{1}{\sqrt{2\pi\sigma^2}^{n-k}} x_i e^{-\frac{|\mathbf{x}_{k+1:n}|^2}{2\sigma^2}} l_\sigma(\mathbf{x}_{1:k})||_{L_1(\mathbb{R}^n)} ||P||_{L_\infty(\mathbb{R}^n)}$$

$$= ||\frac{1}{\sqrt{2\pi\sigma^2}^{n-k}} x_i e^{-\frac{|\mathbf{x}_{k+1:n}|^2}{2\sigma^2}}||_{L_1(\mathbb{R}^{n-k})} ||l_\sigma||_{L_1(\mathbb{R}^k)} ||P||_{L_\infty(\mathbb{R}^n)}$$

Since $|M(\mathbf{x}, \mathbf{y})| \leq \gamma$ for some $\gamma$, we have:

$$|P(\mathbf{x})| \leq \gamma ||\frac{1}{\sqrt{2\pi\sigma^2}^{n-k}} y_j e^{-\frac{|\mathbf{y}_{k+1:n}|^2}{2\sigma^2}} l_\sigma(\mathbf{y}_{1:k})||_{L_1(\mathbb{R}^n)} =$$

$$\gamma ||\frac{1}{\sqrt{2\pi\sigma^2}^{n-k}} e^{-\frac{|\mathbf{y}_{k+1:n}|^2}{2\sigma^2}}||_{L_1(\mathbb{R}^{n-k})} ||y_j l_\sigma(\mathbf{y}_{1:k})||_{L_1(\mathbb{R}^k)} = \gamma ||y_j l_\sigma(\mathbf{y}_{1:k})||_{L_1(\mathbb{R}^k)}$$

Thus,

$$|\langle x_i f_\sigma | M | x_j f_\sigma \rangle| \leq ||\frac{1}{\sqrt{2\pi\sigma^2}^{n-k}} x_i e^{-\frac{|\mathbf{x}_{k+1:n}|^2}{2\sigma^2}}||_{L_1(\mathbb{R}^{n-k})} ||l_\sigma||_{L_1(\mathbb{R}^k)} \gamma ||y_j l_\sigma||_{L_1(\mathbb{R}^k)}$$

Using $||l_\sigma||_{L_1(\mathbb{R}^k)} - ||l||_{L_1(\mathbb{R}^k)} \overset{\sigma \to +0}{\to} 0, ||y_j l_\sigma||_{L_1(\mathbb{R}^k)} - ||y_j l||_{L_1(\mathbb{R}^k)} \overset{\sigma \to +0}{\to} 0$, we see the boundedness of $||l_\sigma||_{L_1(\mathbb{R}^k)} \gamma ||y_j l_\sigma||_{L_1(\mathbb{R}^k)}$ and proceed:

$$\leq C ||\frac{1}{\sqrt{2\pi\sigma^2}^{n-k}} x_i e^{-\frac{|\mathbf{x}_{k+1:n}|^2}{2\sigma^2}}||_{L_1(\mathbb{R}^{n-k})}$$

It is easy to see that $||\frac{1}{\sqrt{2\pi\sigma^2}^{n-k}} x_i e^{-\frac{|\mathbf{x}_{k+1:n}|^2}{2\sigma^2}}||_{L_1(\mathbb{R}^{n-k})} \to 0$ as $\sigma \to 0$, therefore $\langle x_i f_\sigma | M | x_j f_\sigma \rangle \to 0$.

Similarly, we can prove that $\langle x_i f_\sigma | M | x_j f_\sigma \rangle \to 0$ if $i, j > k$.

The entries of the main $k \times k$ minor $[\langle x_i f_\sigma | M | x_j f_\sigma \rangle]_{1 \leq i,j \leq k}$ are bounded, because:

$$\text{Tr } M_{f_\sigma} = \frac{1}{(2\pi\sigma^2)^{n-k}} \iint_{\mathbb{R}^n \times \mathbb{R}^n} \mathbf{x} \cdot \mathbf{y} e^{-\frac{|\mathbf{x}_{k+1:n}|^2}{2\sigma^2}} l_\sigma(\mathbf{x}_{1:k}) M(\mathbf{x}, \mathbf{y}) e^{-\frac{|\mathbf{y}_{k+1:n}|^2}{2\sigma^2}} l_\sigma(\mathbf{y}_{1:k}) d\mathbf{x} d\mathbf{y} \leq$$

$$\frac{\gamma}{(2\pi\sigma^2)^{n-k}} \iint_{\mathbb{R}^n \times \mathbb{R}^n} (|\mathbf{x}_{1:k} \cdot \mathbf{y}_{1:k}| + |\mathbf{x}_{k+1:n} \cdot \mathbf{y}_{k+1:n}|) e^{-\frac{|\mathbf{x}_{k+1:n}|^2 + |\mathbf{y}_{k+1:n}|^2}{2\sigma^2}} l_\sigma(\mathbf{x}_{1:k}) l_\sigma(\mathbf{y}_{1:k}) d\mathbf{x} d\mathbf{y} \leq$$

$$\gamma \iint_{\mathbb{R}^n \times \mathbb{R}^n} |\mathbf{x}_{1:k} \cdot \mathbf{y}_{1:k}| l_\sigma(\mathbf{x}_{1:k}) l_\sigma(\mathbf{y}_{1:k}) d\mathbf{x}_{1:k} d\mathbf{y}_{1:k} + \gamma ||l_\sigma||_{L_1}^2 (n-k)\sigma^2 \leq$$

$$\gamma \sum_{j=1}^{k} ||y_j l_\sigma||_{L_1(\mathbb{R}^k)}^2 + \gamma ||l_\sigma||_{L_1}^2 (n-k)\sigma^2$$

Again, using $||l_\sigma||_{L_1(\mathbb{R}^k)} - ||l||_{L_1(\mathbb{R}^k)} \overset{\sigma \to +0}{\to} 0, ||y_j l_\sigma||_{L_1(\mathbb{R}^k)} - ||y_j l||_{L_1(\mathbb{R}^k)} \overset{\sigma \to +0}{\to} 0$, we obtain the boundedness of RHS. $\qquad \square$

**Corollary 1.** *For any $f \in \mathcal{G}_k$, $\lim_{\sigma \to 0} R(f_\sigma) = 0$.*

*Proof.* W.l.o.g. we can assume that $f = T_l \otimes \delta^{n-k}, l \in \mathcal{S}(\mathbb{R}^k)$. By lemma, all entries of $M_{f_\sigma}$ except those of the main $k \times k$ minor approach 0 as $\sigma \to 0$. This means that

$$\lim_{\sigma \to +0} Q(f_\sigma) = 0$$

where $Q(f_\sigma) = \sum_{i=k+1}^n \langle x_i f_\sigma | M | x_i f_\sigma \rangle$. Let $\mathbf{v}_1, \cdots, \mathbf{v}_n$ be unit eigenvectors of $M_{f_\sigma}$ corresponding to the eigenvalues $\lambda_1 \geq \cdots \geq \lambda_n$, $P = \sum_{i=k+1}^n \mathbf{e}_i \mathbf{e}_i^T$, then

$$R(f_\sigma) = \sum_{i=k+1}^n \lambda_i = \min_{p_i \in [0,1], \sum_1^n p_i = n-k} \sum_{i=1}^n \lambda_i p_i \leq \sum_{i=1}^n \lambda_i \mathrm{Tr}\left(P\mathbf{v}_i\mathbf{v}_i^T P\right) = \mathrm{Tr}\left(PM_{f_\sigma}P\right) = Q(f_\sigma)$$

Since $R(f_\sigma) \leq Q(f_\sigma)$, we obtain $\lim_{\sigma \to 0} R(f_\sigma) = 0$. □

### D.2.1 Proof of Theorem 8

*Proof.* Suppose that a sequence $\{f_i\}_{s=1}^\infty \subseteq \mathcal{S}(\mathbb{R}^n)$ regularly solves (7) and $T \in \varprojlim_{i \to \infty} f_i$. W.l.o.g. we can assume that $T_{f_i} \to^* T$ and $\mathrm{Tr}(M_{f_i})$ is bounded and $I(f_i) + \lambda_i R(f_i) \leq \inf_{f \in \mathcal{S}(\mathbb{R}^n)} I(f) + \lambda_i R(f) + \epsilon_i, \epsilon_i \to 0$. Below we use continuity of $I$ and corollary 1:

$$\inf_{f \in \mathcal{S}(\mathbb{R}^n)} I(f) + \lambda_i R(f) \leq \inf_{f \in \mathcal{G}_k} \inf_{\sigma > 0} I(f_\sigma) + \lambda_i R(f_\sigma) \leq \inf_{f \in \mathcal{G}_k} \lim_{\sigma \to +0} I(f_\sigma) + \lambda_i R(f_\sigma) \leq \inf_{f \in \mathcal{G}_k} I(f)$$

from which we conclude that $\lambda_i R(f_i) \leq \inf_{f \in \mathcal{G}_k} I(f) + \epsilon_i$ and, therefore, $R(f_i) \overset{i \to \infty}{\to} 0$.

For each $l$, let us define $P_l$ as the projection operator to a subspace spanned by first principal components of the matrix $\sqrt{M_{f_l}}$, i.e.

$$P_l = \sum_{i=1}^k \mathbf{v}_i^l \mathbf{v}_i^{l\mathrm{T}}$$

where $\mathbf{v}_1^l, ..., \mathbf{v}_k^l$ are orthonormal eigenvectors that correspond to $k$ largest eigenvalues of $\sqrt{M_{f_l}}$. From the Eckart-Young-Mirsky theorem we see that $R(f_l) = ||\sqrt{M_{f_l}} - P_l\sqrt{M_{f_l}}||_F^2$. Since a set of all projection operators $\{P \in \mathbb{R}^{n \times n} | P^2 = P, P^T = P\}$ is a compact subset of $\mathbb{R}^{n^2}$, one can always find a projection operator $P = \sum_{i=1}^k \mathbf{v}_i \mathbf{v}_i^T$ and a growing subsequence $\{l_s\}$ such that $||P_{l_s} - P||_F \to 0$ as $s \to \infty$. Thus, for the subsequence $\{f_{l_s}\}$ we have:

$$||\sqrt{M_{f_{l_s}}} - P\sqrt{M_{f_{l_s}}}||_F = ||\sqrt{M_{f_{l_s}}} - P_{l_s}\sqrt{M_{f_{l_s}}} + P_{l_s}\sqrt{M_{f_{l_s}}} - P\sqrt{M_{f_{l_s}}}||_F \leq$$

$$||\sqrt{M_{f_{l_s}}} - P_{l_s}\sqrt{M_{f_{l_s}}}||_F + ||P_{l_s} - P||_F ||\sqrt{M_{f_{l_s}}}||_F = \sqrt{R(f_{l_s})} + ||P_{l_s} - P||_F \sqrt{\mathrm{Tr}(M_{f_s})}$$

and using the boundedness of $\mathrm{Tr}(M_{f_s})$ we obtain $||\sqrt{M_{f_{l_s}}} - P\sqrt{M_{f_{l_s}}}||_F \to 0$.

Since $||\sqrt{M_{f_{l_s}}} - P\sqrt{M_{f_{l_s}}}||_F \to 0$, let us complete $\mathbf{v}_1, ..., \mathbf{v}_k$ to an orthonormal basis $\mathbf{v}_1, ..., \mathbf{v}_n$ and make the change of variables $y_i = \mathbf{v}_i^T \mathbf{x}$. Let us denote $V = [\mathbf{v}_1, ..., \mathbf{v}_n]$ and let $V^T = [\mathbf{w}_1, ..., \mathbf{w}_n]$. Then, after that change of variables any function $f(\mathbf{x})$ corresponds to $f'(\mathbf{y}) = f(V\mathbf{y})$ and the kernel $M$ corresponds to $M'(\mathbf{y}, \mathbf{y}') = M(V\mathbf{y}, V\mathbf{y}')$. If we apply that change of variables in the integral expression of $\langle x_i f | M | x_j f \rangle$, we will obtain:

$$\langle x_i f | M | x_j f \rangle = \langle \mathbf{w}_i^T \mathbf{y} f' | M' | \mathbf{w}_j^T \mathbf{y} f' \rangle = \mathbf{w}_i^T \left[\langle y_{i'} f' | M' | y_{j'} f' \rangle\right]_{n \times n} \mathbf{w}_j \Rightarrow$$

$$\mathrm{Re} \langle x_i f | M | x_j f \rangle = \mathbf{w}_i^T \left[\mathrm{Re} \langle y_{i'} f' | M' | y_{j'} f' \rangle\right]_{n \times n} \mathbf{w}_j$$

I.e. $M_f = VM'_{f'}V^T$, or $M'_{f'} = V^T M_f V$. Note that $P = VI_n^k V^T$ where $I_n^k$ is a diagonal matrix whose main $k \times k$ minor is the identity matrix, and all other entries are zeros. Using the fact that the Frobenius norm of orthogonally similar matrices are equal and the identity $V^T\sqrt{M_{f_{l_s}}}V = \sqrt{V^T M_{f_{l_s}}V}$, we obtain:

$$||\sqrt{M_{f_{l_s}}} - P\sqrt{M_{f_{l_s}}}||_F = ||V^T\sqrt{M_{f_{l_s}}}V - V^T P\sqrt{M_{f_{l_s}}}V||_F =$$

$$||\sqrt{V^T M_{f_{l_s}} V} - V^T V I_n^k V^T \sqrt{M_{f_{l_s}}} V||_F = ||\sqrt{M'_{f'_{l_s}}} - I_n^k \sqrt{M'_{f'_{l_s}}}||_F$$

Thus, the property $||\sqrt{M_{f_{l_s}}} - P\sqrt{M_{f_{l_s}}}||_F \to 0$ implies that:

$$\text{Re}\,\langle y_i f'_{l_s} | M' | y_j f'_{l_s} \rangle \to 0, \text{ IF } i > k$$

Moreover, for $i = j$ we have $\text{Re}\,\langle y_i f'_{l_s} | M' | y_j f'_{l_s} \rangle = \langle y_i f'_{l_s} | M' | y_j f'_{l_s} \rangle$. It is easy to see that after the change of variables we still have $f'_{l_s} \to^* T_V$. Since $f'_{l_s} \in \mathcal{S}(\mathbb{R}^n)$, we have $y_i f'_{l_s} \in \mathcal{S}(\mathbb{R}^n)$ and, therefore, $y_i f'_{l_s} \in L_2(\mathbb{R}^n)$. Let us treat now $M'$ as an operator $O(M') : L_2(\mathbb{R}^n) \to L_2(\mathbb{R}^n), O(M')[f](\mathbf{x}) = \int_{\mathbb{R}^n} M'(\mathbf{x}, \mathbf{y}) f(\mathbf{y}) d\mathbf{y}$. Let us take any function $\phi \in L_2(\mathbb{R}^n)$ such that $\psi = O(M')[\phi] \in \mathcal{S}(\mathbb{R}^n)$. Since $O(M')$ is a strictly positive self-adjoint operator, by the Cauchy-Schwarz inequality, we obtain:

$$|\langle y_i f'_{l_s}, O(M')[\phi] \rangle| \le \sqrt{\langle y_i f'_{l_s} | M' | y_i f'_{l_s} \rangle} \sqrt{\langle \phi, O(M')[\phi] \rangle}$$

Therefore, for any $\psi \in \mathcal{R}[O(M')] \cap \mathcal{S}(\mathbb{R}^n)$ and $i > k$ we have $\lim_{s\to\infty} \langle y_i f'_{l_s}, \psi \rangle = \lim_{s\to\infty} \langle f'_{l_s}, y_i \psi \rangle = 0$. Since $f'_{l_s} \to^* T_V$ we obtain $\langle T_V, y_i \psi \rangle = \langle y_i T_V, \psi \rangle = 0$ for any $\psi \in \mathcal{R}[O(M')] \cap \mathcal{S}(\mathbb{R}^n)$. But the denseness of $\mathcal{R}[O(M')] \cap \mathcal{S}(\mathbb{R}^n)$ in $\mathcal{S}(\mathbb{R}^n)$ implies that $y_i T_V = 0$.

Using lemma 2 and $(T_V)_{V^T} = T$ we obtain $T \in \mathcal{G}'_k$. Thus, we proved that $T_{f_i} \to T \in \mathcal{G}'_k$.

Since $I(f_i) \le I(f_i) + \lambda_i R(f_i) \le \inf_{f \in \mathcal{G}'_k} I(f) + \epsilon_i$ and $I$ is continuous, we finally get that $I(T) \le \inf_{f \in \mathcal{G}'_k} I(f)$, i.e. $T \in \text{Arg}\min_{f \in \mathcal{G}'_k} I(f)$. $\square$

### D.2.2 PROOF OF THEOREM 9

*Proof.* Suppose $f^* \in \text{Arg}\min_{f \in \mathcal{G}'_k} I(f) \bigcap \mathcal{B}_k$, i.e. $f^* \in \mathcal{B}_k$ and $I(f^*) = \min_{f \in \mathcal{G}'_k} I(f)$. Since $f^* \in \mathcal{B}_k$, then there exists a sequence $\{s^i\} \subseteq \mathcal{G}_k$ such that $T_{s^i} \to^* f^*$ and $\text{Tr}\,M_{s^i} < \infty$.

Let us define $s^i_\sigma \in \mathcal{S}(\mathbb{R}^n)$ as

$$T_{s^i_\sigma} = T_{s^i} * G^n_\sigma$$

Since $\lim_{\sigma \to 0} R(s^i_\sigma) = 0$ (lemma 4), there exists $\sigma_i > 0$, such that $R(s^i_\sigma) < \frac{1}{i}$ whenever $0 < \sigma \le \sigma_i$. Also, by definition $\text{Tr}\,M_{s^i} = \lim_{\sigma \to 0} \text{Tr}\,M_{s^i_\sigma}$. Therefore, there exists $\sigma'_i > 0$, such that $\text{Tr}\,M_{s^i_\sigma} < \text{Tr}\,M_{s^i} + 1$ whenever $0 < \sigma \le \sigma'_i$.

If we set $\sigma^*_i = \min\{\sigma_i, \sigma'_i, \frac{1}{i}\}$, then a sequence $\{s^i_{\sigma^*_i}\} \subseteq \mathcal{S}(\mathbb{R}^n)$ satisfies:

$$\lim_{i\to\infty} R(s^i_{\sigma^*_i}) = 0$$

$$\text{Tr}\,M_{s^i_{\sigma^*_i}} < \infty$$

and (using lemma 1)

$$T_{s^i_{\sigma^*_i}} \to^* f^*$$

Due to the continuity of $I$ we have

$$\lim_{i\to\infty} I(s^i_{\sigma^*_i}) = I(f^*)$$

Now we set $f_i = s^i_{\sigma^*_i}$, $\lambda_i = \frac{1}{\sqrt{R(f_i)}}$ and we obtain the needed sequence:

$$\lim_{i\to\infty} I(f_i) = \lim_{i\to\infty} I(f_i) + \lambda_i R(f_i) = I(f^*), \lim_{i\to\infty} \lambda_i = +\infty$$

where $\text{Tr}\,M_{f_i}$ is bounded. It remains to check that our sequence regularly solves (7), i.e. $\lim_{i\to\infty} \inf_{f \in \mathcal{S}(\mathbb{R}^n)} I(f) + \lambda_i R(f) = I(f^*)$ (this will imply $\lim_{i\to\infty} I(f_i) + \lambda_i R(f_i) - \inf_{f \in \mathcal{S}(\mathbb{R}^n)} I(f) + \lambda_i R(f) = 0$). The inequality in one direction is obvious,

$$\inf_{f \in \mathcal{S}(\mathbb{R}^n)} I(f) + \lambda_i R(f) \le \inf_{f \in \mathcal{G}_k} \inf_{\sigma > 0} I(f_\sigma) + \lambda_i R(f_\sigma) \le \inf_{f \in \mathcal{G}_k} \lim_{\sigma \to +0} I(f_\sigma) + \lambda_i R(f_\sigma) =$$

$$\inf_{f \in \mathcal{G}_k} I(f) = I(f^*)$$

Let us prove the inverse inequality.

Since $\mathrm{rsol}\,(I(f), R(f)) \neq \emptyset$, there exists $\{\tilde{f}_i\} \subseteq \mathcal{S}(\mathbb{R}^n)$ such that:

$$I(\tilde{f}_i) + \tilde{\lambda}_i R(\tilde{f}_i) \leq \inf_{f \in \mathcal{S}(\mathbb{R}^n)} I(f) + \tilde{\lambda}_i R(f) + \epsilon_i, \ \lim_{s \to +\infty} \tilde{\lambda}_i = +\infty, \ \lim_{i \to +\infty} \epsilon_i = 0, \mathrm{Tr}\, M_{\tilde{f}_i} < \infty$$

and $a = \lim_{i \to +\infty} T_{\tilde{f}_i}$. From theorem 5 we obtain $a \in \mathrm{Arg}\min_{f \in \mathcal{G}'_k} I(f)$.

One can always find a subset $\{\tilde{\lambda}_{d_i}\} \subseteq \{\tilde{\lambda}_i\}$ such that $\tilde{\lambda}_{d_i} < \lambda_i$, $\tilde{\lambda}_{d_i} \to \infty$ and obtain:

$$\inf_{f \in \mathcal{S}(\mathbb{R}^n)} I(f) + \lambda_i R(f) \geq \inf_{f \in \mathcal{S}(\mathbb{R}^n)} I(f) + \tilde{\lambda}_{d_i} R(f) \geq$$

$$I(\tilde{f}_{d_i}) + \tilde{\lambda}_{d_i} R(\tilde{f}_{d_i}) - \epsilon_{d_i} \geq I(\tilde{f}_{d_i}) - \epsilon_{d_i}$$

Therefore,

$$\lim_{i \to \infty} \inf_{f \in \mathcal{S}(\mathbb{R}^n)} I(f) + \lambda_i R(f) \geq \lim_{i \to \infty} I(\tilde{f}_{d_i}) - \epsilon_{d_i} = I(a) = \inf_{f \in \mathcal{G}'_k} I(f) = I(f^*)$$

This proves that $\{f_i\}$ regularly solves (7) and $\lim_{i \to \infty} f_i = f^*$ i.e. $f^* \in \mathrm{rsol}\,(I(f), R(f))$. $\qquad \square$

## E    PROOF OF THEOREM 5

Again we will prove a more general statement.

**Theorem 11.** *If $M$ is a proper and a real-valued kernel, $O(M)$ is bounded and $\mathrm{Tr}\, M_f < \infty$, then $S_f \in \mathcal{B}(L_2^*(\mathbb{R}^n), \mathbb{R}^n)$ and $S_f S_f^\dagger = M_f$. Moreover,*

$$R(f) = \min_{S \in \mathcal{B}(L_2^*(\mathbb{R}^n), \mathbb{R}^n), \mathrm{rank}\, S \leq k} ||S_f - S||_*^2$$

*and the minimum is attained at $S = P_f S_f$ where $P_f = \sum_{i=1}^k \mathbf{u}_i \mathbf{u}_i^\dagger$ and $\{\mathbf{u}_i\}_1^k$ are unit eigenvectors of $M_f$ corresponding to the $k$ largest eigenvalues (counting multiplicities).*

*Proof.* The boundedness of $S_f$ follows from the Cauchy-Schwarz inequality:

$$|S_f[\phi]_i|^2 = |\mathrm{Re}\,\langle \sqrt{O(M)}[x_i f], \phi \rangle|^2 \leq \langle \sqrt{O(M)}[x_i f], \sqrt{O(M)}[x_i f] \rangle \langle \phi, \phi \rangle =$$

$$\langle x_i f, O(M)[x_i f] \rangle \langle \phi, \phi \rangle$$

and therefore:

$$||S_f[\phi]||^2 = \sum_{i=1}^n |S_f[\phi]_i|^2 \leq \mathrm{Tr}\, M_f ||\phi||_{L_2(\mathbb{R}^n)}^2$$

I.e. we have checked that $S_f$ is bounded.

By definition, $S_f^\dagger : \mathbb{R}^n \to L_2^r(\mathbb{R}^n) \times L_2^r(\mathbb{R}^n)$ and $\langle \mathbf{u}, S_f[\phi_1, \phi_2] \rangle = \langle S_f^\dagger[\mathbf{u}], [\phi_1, \phi_2] \rangle, \mathbf{u} \in \mathbb{R}^n, [\phi_1, \phi_2] \in L_2^r(\mathbb{R}^n) \times L_2^r(\mathbb{R}^n)$. Let us denote $f_1 = \mathrm{Re}\, f, f_2 = \mathrm{Im}\, f$. It is easy to see that the following operator satisfies the latter identity:

$$O[\mathbf{u}] = \left[ \sqrt{O(M)}[f_1(\mathbf{x})\mathbf{x}^T\mathbf{u}], \sqrt{O(M)}[f_2(\mathbf{x})\mathbf{x}^T\mathbf{u}] \right]$$

Since the adjoint is unique, then $S_f^\dagger = O$. Let us calculate $S_f S_f^\dagger$:

$$\mathbf{u} \xrightarrow{S_f^\dagger} \left[ \sqrt{O(M)}[f_1(\mathbf{x})\mathbf{x}^T\mathbf{u}], \sqrt{O(M)}[f_2(\mathbf{x})\mathbf{x}^T\mathbf{u}] \right] \xrightarrow{S_f}$$

$$\begin{bmatrix} \langle x_1 f_1(\mathbf{x}), \sqrt{O(M)}[\sqrt{O(M)}[f_1(\mathbf{x})\mathbf{x}^T\mathbf{u}]] \rangle + \langle x_1 f_2(\mathbf{x}), \sqrt{O(M)}[\sqrt{O(M)}[f_2(\mathbf{x})\mathbf{x}^T\mathbf{u}]] \rangle \\ \cdots \\ \langle x_n f_1(\mathbf{x}), \sqrt{O(M)}[\sqrt{O(M)}[f_1(\mathbf{x})\mathbf{x}^T\mathbf{u}]] \rangle + \langle x_n f_2(\mathbf{x}), \sqrt{O(M)}[\sqrt{O(M)}[f_2(\mathbf{x})\mathbf{x}^T\mathbf{u}]] \rangle \end{bmatrix} =$$

$$\begin{bmatrix} \sum_{j=1}^{2} \langle x_1 f_j(\mathbf{x}), O(M)[f_j(\mathbf{x})\mathbf{x}^T \mathbf{u}]\rangle \\ \cdots \\ \sum_{j=1}^{2} \langle x_n f_j(\mathbf{x}), O(M)[f_j(\mathbf{x})\mathbf{x}^T \mathbf{u}]\rangle \end{bmatrix} = [\text{Re} \langle x_i f, M[x_j f]\rangle]_{1 \le i,j \le n} \, \mathbf{u} = M_f \mathbf{u}$$

Thus, $S_f S_f^\dagger = M_f$. Since $\text{Tr}\, S_f S_f^\dagger < \infty$ and $||S_f^\dagger[\mathbf{u}]||^2 \le \langle \mathbf{u}, M_f \mathbf{u}\rangle$, we obtain $S_f^\dagger$ is a bounded operator.

Let $\mathbf{u}_1, \cdots \mathbf{u}_n$ be orthonormal eigenvectors of $\mathcal{M}_f = S_f S_f^\dagger$ and $\lambda_1 \ge \cdots \ge \lambda_{n'} > 0$ be corresponding nonzero eigenvalues. For $\sigma_i = \sqrt{\lambda_i}$ let us define $\mathbf{v}_i = \frac{S_f^\dagger[\mathbf{u}_i]}{\sigma_i}$. Vector $\mathbf{v}_i$ corresponds to a pair of functions:

$$\mathbf{v}_i = \frac{1}{\sigma_i} \left[ \sqrt{O(M)}[f_1(\mathbf{x})\mathbf{x}^T \mathbf{u}_i], \sqrt{O(M)}[f_2(\mathbf{x})\mathbf{x}^T \mathbf{u}_i] \right] \in L_2^r(\mathbb{R}^n) \times L_2^r(\mathbb{R}^n)$$

It is easy to see that $\mathbf{v}_1, \cdots \mathbf{v}_{n'}$ is an orthonormal basis in $\text{Im}\, S_f^\dagger$, and $S_f^\dagger$ can be expanded in the following way:

$$S_f^\dagger = \sum_{i=1}^{n'} \sigma_i \mathbf{v}_i \mathbf{u}_i^\dagger$$

and therefore, SVD for $S_f$ is:

$$S_f = \sum_{i=1}^{n'} \sigma_i \mathbf{u}_i \mathbf{v}_i^\dagger$$

By the Eckart-Young-Mirsky theorem (see Theorem 4.4.7 from Hsing & Eubank (2015)), an optimal $S$ in $\min\limits_{S \in \mathcal{B}(L_2^*(\mathbb{R}^n), \mathbb{R}^n), \text{rank}\, S \le k} ||S_f - S||_*^2$ is defined by a truncation of SVD for $S_f$ at $k$th term, i.e.:

$$S = \sum_{i=1}^{k} \sigma_i \mathbf{u}_i \mathbf{v}_i^\dagger = P_f S_f \tag{16}$$

where $P_f = \sum_{i=1}^{k} \mathbf{u}_i \mathbf{u}_i^\dagger$ is a projection operator to first $k$ principal components of $\mathcal{M}_f$. Moreover, $||S_f - P_f S_f||^2 = \sum_{i=k+1}^{n'} \sigma_i^2 = ||M_f||_{n-k} = R(f)$.

$\square$

## F THE ALTERNATING SCHEME IN THE DUAL SPACE FOR $M(\mathbf{x}, \mathbf{y}) = \zeta(\mathbf{x} - \mathbf{y})$

When $M(\mathbf{x}, \mathbf{y}) = \zeta(\mathbf{x} - \mathbf{y})$, the alternating scheme 1 allows for a reformulation in the dual space. By this we mean that in Scheme 1 we substitute $\hat{\phi}_t$ for the original $\phi_t$. If the primal Scheme 1 deals with operators $S_\phi, S_{\phi_{t-1}}$, the dual version deals with vectors of functions $\sqrt{\hat{\zeta}}\frac{\partial\hat{\phi}}{\partial\mathbf{x}}, \sqrt{\hat{\zeta}}\frac{\partial\hat{\phi}_{t-1}}{\partial\mathbf{x}}$. The substitution is based on the following simple fact:

**Theorem 12.** *If $M(\mathbf{x}, \mathbf{y}) = \zeta(\mathbf{x} - \mathbf{y}), \zeta, \hat{\zeta} \in C(\mathbb{R}^n)$ and $\forall \mathbf{x}\ \hat{\zeta}(\mathbf{x}) > 0$, then there exist constants $c_1$ and $c_2$ such that $||S_\phi - P_{t-1}S_{\phi_{t-1}}||_*^2 = c_1||\ ||\frac{\partial\hat{\phi}}{\partial\mathbf{x}} - P_{t-1}\frac{\partial\hat{\phi}_{t-1}}{\partial\mathbf{x}}||_2\ ||_{L_{2,\hat{\zeta}}(\mathbb{R}^n)}^2$ and $\langle x_i f | M | x_j f \rangle = c_2 \langle \frac{\partial\hat{f}}{\partial x_i}, \frac{\partial\hat{f}}{\partial x_j} \rangle_{L_{2,\hat{\zeta}}(\mathbb{R}^n)}$*

*Proof.* Let $f : \mathbb{R}^n \to \mathbb{C}$ be such that $||x_i f||_{L_2(\mathbb{R}^n)} < \infty$.

$$O(M)[\psi] = \zeta * \psi \Rightarrow \mathcal{F}\{O(M)[\psi]\} \sim \hat{\zeta}\hat{\psi} \Rightarrow \mathcal{F}\left\{\sqrt{O(M)}[\psi]\right\} \sim \sqrt{\hat{\zeta}}\hat{\psi} \Rightarrow$$

$$S_f[\psi]_i = \text{Re}\langle x_i f, \sqrt{O(M)}[\psi]\rangle \sim \text{Re}\langle \mathcal{F}\{x_i f\}, \mathcal{F}\left\{\sqrt{O(M)}[\psi]\right\}\rangle \sim$$

$$\text{Re}\langle \text{i}\frac{\partial\hat{f}}{\partial x_i}, \sqrt{\hat{\zeta}}\hat{\psi}\rangle = \text{Re}\langle \text{i}\sqrt{\hat{\zeta}}\frac{\partial\hat{f}}{\partial x_i}, \hat{\psi}\rangle$$

Since $S_f[\psi]_i = \text{Re} \langle (S_f)_i, \psi \rangle \sim \text{Re} \langle \widehat{(S_f)_i}, \hat{\psi} \rangle$, we obtain

$$\widehat{(S_f)_i} = \kappa \sqrt{\hat{\zeta}} \frac{\partial \hat{f}}{\partial x_i} \tag{17}$$

where $\kappa$ is a constant.

Let us now introduce a vector of functions $V_f = [(S_f)_1, \cdots, (S_f)_n]^T \in L_2^n(\mathbb{R}^n)$. Using 17 we obtain $\widehat{(S_f)_i} = \kappa \sqrt{\hat{\zeta}} \frac{\partial \hat{f}}{\partial x_i}$, and therefore:

$$\widehat{V}_f = \kappa \sqrt{\hat{\zeta}} \frac{\partial \hat{f}}{\partial \mathbf{x}}$$

Thus, the expression $||S_\phi - P_{t-1} S_{\phi_{t-1}}||_*^2$ in the alternating scheme can be rewritten as:

$$||V_\phi - P_{t-1} V_{\phi_{t-1}}||_{L_2^n(\mathbb{R}^n)}^2 \sim ||\kappa \sqrt{\hat{\zeta}} \frac{\partial \hat{\phi}}{\partial \mathbf{x}} - P_{t-1} \kappa \sqrt{\hat{\zeta}} \frac{\partial \hat{\phi}_{t-1}}{\partial \mathbf{x}}||_{L_2^n(\mathbb{R}^n)}^2 \sim$$

$$|| \,||\frac{\partial \hat{\phi}}{\partial \mathbf{x}} - P_{t-1} \frac{\partial \hat{\phi}_{t-1}}{\partial \mathbf{x}}||_2 \,||_{L_{2,\hat{\zeta}}(\mathbb{R}^n)}^2$$

The matrix $M_f$ can also be calculated from $\hat{f}$ using the following identity:

$$\langle x_i f, M[x_j f] \rangle = \langle x_i f, \zeta * (x_j f) \rangle \sim \langle \frac{\partial \hat{f}}{\partial x_i}, \hat{\zeta} \frac{\partial \hat{f}}{\partial x_j} \rangle = \langle \frac{\partial \hat{f}}{\partial x_i}, \frac{\partial \hat{f}}{\partial x_j} \rangle_{L_{2,\hat{\zeta}}(\mathbb{R}^n)}$$

$\square$

Let us introduce a function $\tilde{I}$ such that $\tilde{I}(f) = I(\hat{f})$. Then, we see that all steps in Scheme 1 can be performed with $\hat{\phi}_t$ rather than with $\phi_t$, using the algorithm 2.

Informally, the dual algorithm works as follows: at each iteration $t$ we compute a function $\hat{\phi}_t$ adapting it to data (the term $\tilde{I}(\hat{\phi})$) and adapting its gradient field to the rank reduced gradient field of the previous $\hat{\phi}_{t-1}$. For a sufficiently large $T$, it will converge and $\hat{\phi}_T \approx \hat{\phi}_{T-1}$. Then, the second term in the last step will be approximately equal to $\lambda || \,||\frac{\partial \hat{\phi}_T}{\partial \mathbf{x}} - P_{T-1} \frac{\partial \hat{\phi}_T}{\partial \mathbf{x}}||_2 \,||_{L_{2,\hat{\zeta}}(\mathbb{R}^n)}^2$, enforcing $\frac{\partial \hat{\phi}_T}{\partial \mathbf{x}} \approx P_{T-1} \frac{\partial \hat{\phi}_T}{\partial \mathbf{x}}$ for random $\mathbf{x} \sim \frac{\hat{\zeta}}{||\hat{\zeta}||_{L_1}}$. Thus, gradients $\frac{\partial \hat{\phi}_T}{\partial \mathbf{x}}$ lie in a $k$-dimensional subspace $\text{col}\, P_{T-1}$. This last property is a characteristic property of functions from $\mathcal{F}_k$.

---

**Algorithm 2** The alternating scheme in the dual space.

$P_0 \longleftarrow \mathbf{0}, \hat{\phi}_0 \longleftarrow \mathbf{0}$
**for** $t = 1, \cdots, T$ **do**
  $\hat{\phi}_t \leftarrow \arg\min_{\hat{\phi}} \tilde{I}(\hat{\phi}) + \tilde{\lambda} || \,||\frac{\partial \hat{\phi}}{\partial \mathbf{x}} - P_{t-1} \frac{\partial \hat{\phi}_{t-1}}{\partial \mathbf{x}}||_2 \,||_{L_{2,\hat{\zeta}}(\mathbb{R}^n)}^2$
  Calculate $M_t = \left[ \text{Re} \langle \frac{\partial \hat{\phi}_t}{\partial x_i}, \frac{\partial \hat{\phi}_t}{\partial x_j} \rangle_{L_{2,\hat{\zeta}}(\mathbb{R}^n)} \right]$
  Find $\{\mathbf{v}_i\}_1^n$ s.t. $M_t \mathbf{v}_i = \lambda_i \mathbf{v}_i, \lambda_1 \geq \cdots \geq \lambda_n$
  $P_t \longleftarrow \sum_{i=1}^k \mathbf{v}_i \mathbf{v}_i^T$
**Output:** $\mathbf{v}_1, \cdots, \mathbf{v}_k$

---

# G   A NUMERICAL ALTERNATING SCHEME FOR MMD

## G.1   STRUCTURE OF $\mathcal{F}[\mathcal{P}_k]$

From theorems 1 and 2, $\mathcal{F}[\mathcal{P}_k] \subseteq \overline{\mathcal{F}_k}^*$. In fact, a famous theorem of Bochner (1932) gives us that the Fourier transform of any positive finite Borel measure is a continuous positive definite function. That is, if $f \in \mathcal{F}[\mathcal{P}]$, then for any distinct $\mathbf{y}_1, \cdots, \mathbf{y}_s \in \mathbb{R}^n$ the matrix $[f(\mathbf{y}_i - \mathbf{y}_j)]_{i,j=\overline{1,n}}$ is

positive semidefinite. Since $\mu(\mathbb{R}^n) = 1$, we additionally have $f(\mathbf{0}) = 1$. Let PDF denote the set of all continuous positive definite functions on $\mathbb{R}^n$ and

$$\mathcal{M}_k = \{f \in \text{PDF} | \exists \mathbf{v}_1, ..., \mathbf{v}_k \in \mathbb{R}^n, g : \mathbb{R}^k \to \mathbb{C} \text{ s.t. } f(\mathbf{x}) = g(\mathbf{v}_1^T \mathbf{x}, ..., \mathbf{v}_k^T \mathbf{x}), f(\mathbf{0}) = 1\} \quad (18)$$

Thus, the following characterization of $\mathcal{F}[\mathcal{P}_k]$ becomes evident.

**Theorem 13.** $\mathcal{F}[\mathcal{P}_k] = \mathcal{M}_k$.

## G.2 THE DUAL FORM OF MMD

Let us define another gaussian kernel $\gamma(\mathbf{x}) = e^{-\frac{h^2 |\mathbf{x}|^2}{2}} = \mathcal{F}[k]$. Let $p_{\text{data}}(\mathbf{x})$ denote the characteristic function of the random vector $\mathbf{X}_{\text{data}} \sim \mu_{\text{data}}$. By definition, $p_{\text{data}}(\mathbf{x}) = \mathbb{E}[e^{i\mathbf{X}_{\text{data}}^T \mathbf{x}}] = \frac{1}{N}\sum_{i=1}^N e^{i\mathbf{x}_i^T \mathbf{x}}$. Thus, $p_{\text{data}} \propto \mathcal{F}^{-1}[\mu_{\text{data}}]$ and $\mu_{\text{data}} \propto \mathcal{F}[p_{\text{data}}]$.

Using the isometry property of the Fourier transform for $L_2(\mathbb{R}^n)$ and the convolution theorem, we see that:

$$d_{\text{MMD}}(\mu, \nu) = ||k * \mu - k * \nu||_{L_2(\mathbb{R}^n)} \propto ||\gamma(\mathbf{x})(\mathcal{F}[\mu](\mathbf{x}) - \mathcal{F}[\nu](\mathbf{x}))||_{L_2(\mathbb{R}^n)}$$

Thus, from Theorem 13 we obtain that the task 6 is equivalent to:

$$||p_{\text{data}} - q||_{L_{2,\gamma^2}(\mathbb{R}^n)} \to \min_{q \in \mathcal{M}_k} \quad (19)$$

## G.3 ALGORITHMS FOR MMD

Let $\Pi_k : \mathcal{G}_k \to \{1, +\infty\}$ and $\mathbf{M}_k : \mathcal{F}_k \to \{1, +\infty\}$ be simple penalty functions:

$$\Pi_k(\phi) = 1, \text{ if } \phi \in \mathcal{P}_k \text{ and } \Pi_k(\phi) = \infty, \text{ otherwise}$$

$$\mathbf{M}_k(\phi) = 1, \text{ if } \phi \in \mathcal{M}_k \text{ and } \mathbf{M}_k(\phi) = \infty, \text{ otherwise}$$

Then, the task 6 is equivalent to:

$$I(\phi) = d_{\text{MMD}}^2(\mu_{\text{data}}, \phi)\Pi_k(\phi) \to \inf_{\phi \in \mathcal{G}_k}$$

From the result of the previous section we see that if $I(\phi) = \tilde{I}(\hat{\phi})$, then:

$$\tilde{I}(\hat{\phi}) = ||p_{\text{data}} - \hat{\phi}||_{L_{2,\gamma^2}(\mathbb{R}^n)}^2 \mathbf{M}_k(\hat{\phi})$$

Thus, the Algorithm 3 is an adaptation of Algorithm 2 to MMD.

---

**Algorithm 3** The alternating scheme in the dual space for MMD

---

$P_0 \longleftarrow \mathbf{0}, q_0 \longleftarrow \mathbf{0}$
**for** $t = 1, \cdots, T$ **do**
   1 $q_t \longleftarrow \arg\min_{q \in \mathcal{M}_k} \int_{\mathbb{R}^n} \gamma(\mathbf{x})^2 |p_{\text{data}}(\mathbf{x}) - q(\mathbf{x})|^2 d\mathbf{x} + \lambda \int_{\mathbb{R}^n} \hat{\zeta}(\mathbf{x})||\frac{\partial q}{\partial \mathbf{x}} - P_{t-1}\frac{\partial q_{t-1}}{\partial \mathbf{x}}||_2^2 d\mathbf{x}$
   2 Calculate $M_t = \left[\langle\frac{\partial q_t}{\partial x_i}, \frac{\partial q_t}{\partial x_j}\rangle_{L_{2,\hat{\zeta}}(\mathbb{R}^n)}\right]$
   3 Find $\{\mathbf{v}_i\}_1^n$ s.t. $M_t\mathbf{v}_i = \lambda_i\mathbf{v}_i, \lambda_1 \geq \cdots \geq \lambda_n$
   4 $P_t \longleftarrow \sum_{i=1}^k \mathbf{v}_i\mathbf{v}_i^T$
**Output:** $\mathcal{L} = span(\mathbf{v}_1, \cdots, \mathbf{v}_k)$

---

If the function $p_{\text{data}}$ is real-valued, then only real-valued functions can appear in the Algorithm 3. This assumption can be satisfied by adding reflections of initial points to the dataset (after it was centered).

At step 1, we search over $q$ given in the following parameterized form:

$$q_\theta(\mathbf{x}) = \sum_{i=1}^{\text{nn}} \alpha_i \cos(\omega_i^T \mathbf{x}) \quad (20)$$

where $\alpha_i > 0$ and $\sum_{i=1}^{nn} \alpha_i = 1$. In our implementation, we set $[\alpha_i]_{i=\overline{1,nn}} = \text{softmax}([u_i]_{i=\overline{1,nn}})$ and $u_i$'s are unconstrained. The number of neurons in a single layer neural network with a cosine activation function, nn, is a hyperparameter. Let us denote parameters $\{\omega_i, u_i\}_{i=1}^{nn}$ by $\theta$. It is easy to see the function $q_\theta$ is positive definite. Moreover, using Theorem 2 from Barron (1993), it can be shown that a set of all such functions, i.e. the convex hull of $\{cos(\omega^T \mathbf{x}) | \omega \in \mathbb{R}^n\}$, is dense in a set of real-valued functions from $\mathcal{M}_k$. Though this parameterization is quite natural, finding architectures with more expressive power in a space of real-valued positive definite functions is an open problem.

Now, to minimize

$$\Psi(\theta) = \int_{\mathbb{R}^n} \gamma(\mathbf{x})^2 |p_{\text{data}}(\mathbf{x}) - q_\theta(\mathbf{x})|^2 d\mathbf{x} + \lambda \int_{\mathbb{R}^n} \hat{\zeta}(\mathbf{x}) ||\frac{\partial q_\theta}{\partial \mathbf{x}} - P_{t-1} \frac{\partial q_{\theta_{t-1}}}{\partial \mathbf{x}}||_2^2 d\mathbf{x}$$

with stochastic gradient descent methods (in our case, the Adam optimizer) we need to have an unbiased estimator of

$$\nabla_\theta \Psi(\theta) \propto \mathbb{E}_{\mathbf{z} \sim \gamma^2} \nabla_\theta |p_{\text{data}}(\mathbf{z}) - q_\theta(\mathbf{z})|^2 + \tilde{\lambda} \mathbb{E}_{\mathbf{z}' \sim \hat{\zeta}} \nabla_\theta ||\frac{\partial q_\theta}{\partial \mathbf{x}}(\mathbf{z}') - P_{t-1} \frac{\partial q_{\theta_{t-1}}}{\partial \mathbf{x}}(\mathbf{z}')||_2^2$$

where $\mathbf{z} \sim f$ denotes that the random vector $\mathbf{z}$ is sampled according to the probability density function $\frac{f(\mathbf{x})}{\int_{\mathbb{R}^n} f(\mathbf{x}) d\mathbf{x}}$. Thus, a natural estimator of the gradient is:

$$\frac{1}{m} \sum_{i=1}^m \nabla_\theta |p_{\text{data}}(\mathbf{z}_i) - q_\theta(\mathbf{z}_i)|^2 + \frac{\tilde{\lambda}}{m} \sum_{i=1}^m \nabla_\theta ||\frac{\partial q_\theta(\boldsymbol{\xi}_i))}{\partial \mathbf{x}} - P_{t-1} \frac{\partial q_{\theta_{t-1}}(\boldsymbol{\xi}_i))}{\partial \mathbf{x}}||_2^2$$

where $\{\mathbf{z}_i\}_{i=1}^m \sim^{iid} \gamma^2$ and $\{\boldsymbol{\xi}_i\}_{i=1}^m \sim^{iid} \hat{\zeta}$.

The last important issue with the practical numerical algorithm is the calculation of $M_t$ at step 2. It is easy to see that:

$$M_t = \mathbb{E}_{\boldsymbol{\chi} \sim \hat{\zeta}} \frac{\partial q_t}{\partial \mathbf{x}}(\boldsymbol{\chi}) \frac{\partial q_t}{\partial \mathbf{x}}(\boldsymbol{\chi})^T$$

In practice we sample $\boldsymbol{\chi}_1, \cdots, \boldsymbol{\chi}_l \sim \hat{\zeta}$ and estimate $M_t$ as follows:

$$M_t \approx \frac{1}{l} \sum_{i=1}^l \frac{\partial q_t}{\partial \mathbf{x}}(\boldsymbol{\chi}_i) \frac{\partial q_t}{\partial \mathbf{x}}(\boldsymbol{\chi}_i)^T$$

The details of the numerical algorithm are given below 4. In all our experiments with MMD we set $\hat{\zeta} = \gamma^2$.

---

**Algorithm 4** The numerical algorithm for MMD. Hyperparameters: $\tilde{\lambda}, h, \sigma, m, l, \alpha, \beta_1, \beta_2, nn$.

---

$P_0 \longleftarrow \mathbf{0}, \theta_0 \longleftarrow \mathbf{0}$
**for** $t = 1, \cdots, T$ **do**
    **while** $\theta$ has not converged **do**
        Sample $\{\mathbf{z}_i\}_{i=1}^m \sim^{iid} \gamma^2$
        Sample $\{\boldsymbol{\xi}_i\}_{i=1}^m \sim^{iid} \hat{\zeta}$
        $L \longleftarrow \frac{1}{m} \sum_{i=1}^m |p_{\text{data}}(\mathbf{z}_i) - q_\theta(\mathbf{z}_i)|^2 + \frac{\tilde{\lambda}}{m} \sum_{i=1}^m ||\frac{\partial q_\theta(\boldsymbol{\xi}_i))}{\partial \mathbf{x}} - P_{t-1} \frac{\partial q_{\theta_{t-1}}(\boldsymbol{\xi}_i))}{\partial \mathbf{x}}||_2^2$
        $\theta \longleftarrow \text{Adam}(\nabla_\theta L, \theta, \alpha, \beta_1, \beta_2)$
    $\theta_t \longleftarrow \theta$
    Sample $\{\boldsymbol{\chi}_i\}_{i=1}^l \sim^{iid} \hat{\zeta}$
    Calculate $M_t = \frac{1}{l} \sum_{i=1}^l \frac{\partial q_{\theta_t}(\boldsymbol{\chi}_i))}{\partial \mathbf{x}} \frac{\partial q_{\theta_t}(\boldsymbol{\chi}_i))}{\partial \mathbf{x}}^T$
    Find $\{\mathbf{v}_i\}_1^n$ s.t. $M_t \mathbf{v}_i = \lambda_i \mathbf{v}_i, \lambda_1 \geq \cdots \geq \lambda_n$
    $P_t \longleftarrow \sum_{i=1}^k \mathbf{v}_i \mathbf{v}_i^T$
**Output:** $\mathbf{v}_1, \cdots, \mathbf{v}_k$

---

## H  A NUMERICAL ALTERNATING SCHEME FOR HM

### H.1  THE DUAL FORM OF HM

Due to a well-known relationship between moments of the probability measure $\mu$ and its characteristic function $p$, i.e. $\mathrm{i}^s m_{i_1 \cdots i_s} = \frac{\partial^s p(\mathbf{0})}{\partial x_{i_1} \cdots \partial x_{i_s}}$, the task 7 is equivalent to:

$$\sum_{s=1}^{4} \frac{\lambda_s}{n^s} \sum_{1 \le i_1, \cdots, i_s \le n} |\frac{\partial^s p_{\mathrm{data}}(\mathbf{0})}{\partial x_{i_1} \cdots \partial x_{i_s}} - \frac{\partial^s q(\mathbf{0})}{\partial x_{i_1} \cdots \partial x_{i_s}}|^2 \to \min_{q \in \mathcal{M}_k} \tag{21}$$

Note that the maximum mean discrepancy distance and the distance based on higher moments are substantially different. Indeed, even if we set $h$ as a large value (which makes $\frac{1}{h} \approx 0$), the MMD distance, unlike the HM distance, neglects higher order derivatives of the characteristic functions in the neigbourhood of the origin. Moreover, from the dual form 21 it is clear that $d_{\mathrm{HM}}(\mu_{\mathrm{data}}, \nu)$ is a degenerate case of a weighted Sobolev norm between characteristic functions of $\mu_{\mathrm{data}}$ and $\nu$.

### H.2  ALGORITHMS FOR HM

Analogously to the case of MMD we see that the task 7 is equivalent to:

$$I(\phi) = d_{\mathrm{HM}}(\mu_{\mathrm{data}}, \phi)^2 \Pi_k(\phi) \to \inf_{\phi \in \mathcal{G}_k}$$

and

$$\tilde{I}(\hat{\phi}) = \sum_{s=1}^{4} \frac{\lambda_s}{n^s} \sum_{1 \le i_1, \cdots, i_s \le n} |\frac{\partial^s p_{\mathrm{data}}(\mathbf{0})}{\partial x_{i_1} \cdots \partial x_{i_s}} - \frac{\partial^s \hat{\phi}(\mathbf{0})}{\partial x_{i_1} \cdots \partial x_{i_s}}|^2 \mathbf{M}_k(\hat{\phi})$$

Thus, the Algorithm 5 is an adaptation of Algorithm 2 to HM.

---

**Algorithm 5** The alternating scheme in the dual space for HM

$P_0 \longleftarrow \mathbf{0}, q_0 \longleftarrow \mathbf{0}$
**for** $t = 1, \cdots, T$ **do**
    1 $q_t \longleftarrow \arg\min_{q \in \mathcal{M}_k} \sum_{s=1}^{4} \frac{\lambda_s}{n^s} \sum_{1 \le i_1, \cdots, i_s \le n} |\frac{\partial^s p_{\mathrm{data}}(\mathbf{0})}{\partial x_{i_1} \cdots \partial x_{i_s}} - \frac{\partial^s q(\mathbf{0})}{\partial x_{i_1} \cdots \partial x_{i_s}}|^2 + \lambda \int_{\mathbb{R}^n} \hat{\zeta}(\mathbf{x}) || \frac{\partial q}{\partial \mathbf{x}} -$
$P_{t-1} \frac{\partial q_{t-1}}{\partial \mathbf{x}} ||_2^2 d\mathbf{x}$
    2 Calculate $M_t = \left[ \langle \frac{\partial q_t}{\partial x_i}, \frac{\partial q_t}{\partial x_j} \rangle_{L_{2,\hat{\zeta}}(\mathbb{R}^n)} \right]$
    3 Find $\{\mathbf{v}_i\}_1^n$ s.t. $M_t \mathbf{v}_i = \lambda_i \mathbf{v}_i, \lambda_1 \ge \cdots \ge \lambda_n$
    4 $P_t \longleftarrow \sum_{i=1}^{k} \mathbf{v}_i \mathbf{v}_i^T$
**Output:** $\mathcal{L} = span(\mathbf{v}_1, \cdots, \mathbf{v}_k)$

---

Again, as in a numerical algorithm for MMD, at step 1, we search over $q$ given in the form 20. The objective of step 1 can be represented as:

$$\Phi(\theta) = \sum_{s=1}^{4} \lambda_s \mathbb{E}_{i_1, \cdots, i_s \sim^{iid} \mathcal{U}(1,n)} |\frac{\partial^s (p_{\mathrm{data}} - q_\theta)(\mathbf{0})}{\partial x_{i_1} \cdots \partial x_{i_s}}|^2 + \tilde{\lambda} \mathbb{E}_{\mathbf{z}' \sim \hat{\zeta}} || \frac{\partial q_\theta}{\partial \mathbf{x}}(\mathbf{z}') - P_{t-1} \frac{\partial q_{\theta_{t-1}}}{\partial \mathbf{x}}(\mathbf{z}') ||_2^2$$

where $\mathcal{U}(1, n)$ is the discrete uniform distribution over $\{1, \cdots, n\}$. To apply the stochastic gradient descent methods we need to have an unbiased estimator of $\nabla_\theta \Phi(\theta)$ which is equal to:

$$\sum_{s=1}^{4} \lambda_s \mathbb{E}_{i_1, \cdots, i_s \sim^{iid} \mathcal{U}(1,n)} \nabla_\theta |\frac{\partial^s (p_{\mathrm{data}} - q_\theta)(\mathbf{0})}{\partial x_{i_1} \cdots \partial x_{i_s}}|^2 + \tilde{\lambda} \mathbb{E}_{\mathbf{z}' \sim \hat{\zeta}} \nabla_\theta || \frac{\partial q_\theta}{\partial \mathbf{x}}(\mathbf{z}') - P_{t-1} \frac{\partial q_{\theta_{t-1}}}{\partial \mathbf{x}}(\mathbf{z}') ||_2^2$$

Thus, a natural estimator of the gradient is:

$$\sum_{s=1}^{4} \frac{\lambda_s}{m_1} \sum_{i=1}^{m_1} \nabla_\theta |\frac{\partial^s (p_{\mathrm{data}} - q_\theta)(\mathbf{0})}{\partial x_{a[s,i,1]} \partial x_{a[s,i,2]} \cdots \partial x_{a[s,i,s]}}|^2 + \frac{\tilde{\lambda}}{m_2} \sum_{i=1}^{m_2} \nabla_\theta || \frac{\partial q_\theta(\boldsymbol{\xi}_i)}{\partial \mathbf{x}} - P_{t-1} \frac{\partial q_{\theta_{t-1}}(\boldsymbol{\xi}_i)}{\partial \mathbf{x}} ||_2^2$$

where $\{a[s,i,j]\}_{s=\overline{1,4}, i=\overline{1,m_1}, j=\overline{1,s}} \sim^{iid} \mathcal{U}(1,n)$ and $\{\boldsymbol{\xi}_i\}_{i=1}^{m_2} \sim^{iid} \hat{\zeta}$. Overall, we obtain the following Algorithm 6.

---

**Algorithm 6** The numerical algorithm for HM. Hyperparameters: $\tilde{\lambda}, \{\lambda_s\}_{s=\overline{1,4}}, m_1, m_2, l, \alpha, \beta_1, \beta_2, \text{nn}.$

---

$P_0 \longleftarrow \mathbf{0}, \theta_0 \longleftarrow \mathbf{0}$
**for** $t = 1, \cdots, T$ **do**
    **while** $\theta$ has not converged **do**
        Sample $\{a[s,i,j]\}_{s=\overline{1,4}, i=\overline{1,m_1}, j=\overline{1,s}} \sim^{iid} \mathcal{U}(1, n)$
        Sample $\{\boldsymbol{\xi}_i\}_{i=1}^{m_2} \sim^{iid} \hat{\zeta}$
        $L \longleftarrow \sum_{s=1}^{4} \frac{\lambda_s}{m_1} \sum_{i=1}^{m_1} \nabla_\theta |\frac{\partial^s (p_{\text{data}} - q_\theta)(\mathbf{0})}{\partial x_{a[s,i,1]} \partial x_{a[s,i,2]} \cdots \partial x_{a[s,i,s]}}|^2 + \frac{\tilde{\lambda}}{m_2} \sum_{i=1}^{m_2} \nabla_\theta \| \frac{\partial q_\theta(\boldsymbol{\xi}_i))}{\partial \mathbf{x}} -$
        $P_{t-1} \frac{\partial q_{\theta_{t-1}}(\boldsymbol{\xi}_i))}{\partial \mathbf{x}} \|_2^2$
        $\theta \longleftarrow \text{Adam}(\nabla_\theta L, \theta, \alpha, \beta_1, \beta_2)$
    $\theta_t \longleftarrow \theta$
    Sample $\{\boldsymbol{\chi}_i\}_{i=1}^{l} \sim^{iid} \hat{\zeta}$
    Calculate $M_t = \frac{1}{l} \sum_{i=1}^{l} \frac{\partial q_{\theta_t}(\boldsymbol{\chi}_i))}{\partial \mathbf{x}} \frac{\partial q_{\theta_t}(\boldsymbol{\chi}_i))}{\partial \mathbf{x}}^T$
    Find $\{\mathbf{v}_i\}_1^n$ s.t. $M_t \mathbf{v}_i = \lambda_i \mathbf{v}_i, \lambda_1 \geq \cdots \geq \lambda_n$
    $P_t \longleftarrow \sum_{i=1}^{k} \mathbf{v}_i \mathbf{v}_i^T$
**Output:** $\mathbf{v}_1, \cdots, \mathbf{v}_k$

---

## I A NUMERICAL ALTERNATING SCHEME FOR WD

By Theorem 6, the task 13 is equivalent to $\min_{\mu \in \mathcal{P}_k} W(\mu, \mu_{\text{data}})$, or to the following task:

$$I(\phi) \to \inf_{\phi \in \mathcal{G}_k}$$

where $I(T_\mu) = W(\mu, \mu_{\text{data}})$ if $\mu \in \mathcal{P}_k$ and $I(\phi) = \infty$, if otherwise. The alternating scheme 1 is designed to solve the penalty form of the problem, i.e.

$$I(\phi) + \lambda R(\phi) \to \min_{\phi \in \mathcal{S}(\mathbb{R}^n)}$$

which is equivalent to

$$W(\phi, \mu_{\text{data}}) + \lambda R(\phi) \to \min_{\phi \in \mathcal{S}_p(\mathbb{R}^n)}$$

where $\mathcal{S}_p(\mathbb{R}^n) \subseteq \mathcal{S}(\mathbb{R}^n)$ is a set of Schwartz functions that can serve as pdf: $\phi(\mathbf{x}) \geq 0$, $\int_{\mathbb{R}^n} \phi(\mathbf{x}) d\mathbf{x} = 1$. A numerical version of the alternating scheme requires additional specifications on: a) how to minimize over $\phi$ at step 1, and b) how to estimate $M_{\phi_t}$.

### I.1 HOW TO MINIMIZE OVER $\phi$?

In the case of WD, the minimization step of the alternating scheme makes the following:

$$\phi_t \longleftarrow \arg \min_{\phi \in \mathcal{S}_p(\mathbb{R}^n)} W(\phi, \mu_{\text{data}}) + \lambda \|S_\phi - P_{t-1} S_{\phi_{t-1}}\|^2 \tag{22}$$

where $S_f = \sqrt{O(M)}[\mathbf{x} f(\mathbf{x})]$.

For a numerical implementation of that step we need to choose some family of functions that is dense in $\mathcal{S}_p(\mathbb{R}^n)$ (or, rich enough to approach the solution $\mu^*$). Following the tradition of GAN research let us assume that the family is given in the following form[2]:

$$\mathcal{H} = \{\phi_\theta | \phi_\theta(\mathbf{x}) \text{ is pdf of random vector } g_\theta(\mathbf{z}), \mathbf{z} \sim p(\mathbf{z}), \theta \in \Theta\} \tag{23}$$

where $\{g_\theta | \theta \in \Theta\}$ is a parameterized family of smooth functions (usually, a neural network) and $p(\mathbf{z})$ is some fixed distribution (usually, the gaussian distribution). Following Arjovsky et al. (2017), we make the assumption 1. In a numerical algorithm we need an access to a procedure that samples according to $\phi_\theta(\mathbf{x})$, not the function itself.

**Assumption 1.** $\|g_{\theta'}(\mathbf{z}') - g_\theta(\mathbf{z})\| \leq L(\theta, \mathbf{z})(\|\theta' - \theta\| + \|\mathbf{z}' - \mathbf{z}\|)$ *where*

$$\mathbb{E}_{\mathbf{z} \sim p(\mathbf{z})} L(\theta, \mathbf{z}) < +\infty$$

---

[2]If $\mathcal{H} \subseteq \mathcal{S}(\mathbb{R}^n)$ is not satisfied, then we can choose $\mathcal{H}_\epsilon = \{\phi_\theta * G_\epsilon^n | \theta \in \Theta\}$ for a very small $\epsilon$.

Thus, instead of solving 22 we solve:

$$\phi_t \longleftarrow \arg\min_{\phi \in \mathcal{H}} W(\phi, \mu_{\mathrm{data}}) + \lambda ||S_\phi - P_{t-1} S_{\phi_{t-1}}||^2$$

taking into account that $\phi_{t-1} \in \mathcal{H}$.

The Kantorovich-Rubinstein duality theorem gives us that:

$$W(\phi_\theta, \mu_{\mathrm{data}}) = \max_{f:||f_{\mathbf{x}}|| \leq 1} \mathbb{E}_{\mathbf{x} \sim \mu_{\mathrm{data}}}[f(\mathbf{x})] - \mathbb{E}_{\mathbf{z} \sim p(\mathbf{z})}[f(g_\theta(\mathbf{z}))]$$

which turns 22 into the following minimax task:

$$\phi_t \longleftarrow \arg\min_{\phi \in \mathcal{H}} \max_{f:||f_{\mathbf{x}}|| \leq 1} \mathbb{E}_{\mathbf{x} \sim \mu_{\mathrm{data}}}[f(\mathbf{x})] - \mathbb{E}_{\mathbf{z} \sim p(\mathbf{z})}[f(g_\theta(\mathbf{z}))] + \lambda ||S_\phi - P_{t-1} S_{\phi_{t-1}}||^2 \quad (24)$$

In practice, we choose a family of functions $\mathcal{L} = \{f_w | w \in \mathcal{W}\}$ and internal maximization is made over $w \in \mathcal{W}$ with an additional penalty term that penalizes a violation of the Lipschitz condition: $\forall \mathbf{x} : ||f_{\mathbf{x}}|| \leq 1$.

A family of minimax algorithms for the minimization of $W(\phi_\theta, \mu_{\mathrm{emp}})$ was developed in a series of papers Arjovsky et al. (2017); Gulrajani et al. (2017); Wei et al. (2018). The standard minimax scheme that gained popularity in GAN literature iterates two steps: a) $n_{\mathrm{iter}}$ times make a gradient ascent over $w \in \mathcal{W}$, b) make a gradient descent over $\theta$. The task 24 can be viewed as a Wasserstein GAN with an additional regularization term $\lambda T(\theta)$ where $T(\theta) = ||S_{\phi_\theta} - P_{t-1} S_{\phi_{\theta_{t-1}}}||^2$. To adapt these algorithms to the minimization of our function, we only need to have an unbiased estimator of the gradient $\frac{\partial T}{\partial \theta}$. This estimator is needed for the generator to make its gradient descent step. The discriminator's part of the algorithm (in which we maximize over Lipschitz functions $f_w$) can be set in a standard fashion — we choose Petzka et al. (2018)'s version, in which the term $\max\{0, ||\frac{\partial f_w}{\partial \mathbf{x}}(\xi \mathbf{x} + (1 - \xi) g_\theta(\mathbf{z}))|| - 1\}^2$ enforces Lipschitz condition (see step (*) of the Algorithm 7).

---

**Algorithm 7** Numerical algorithm for WD. We use $M(\mathbf{x}, \mathbf{y}) = e^{-\frac{||\mathbf{x}-\mathbf{y}||^2}{n}}$ and default values of $\lambda = 10, \Lambda = 100, n_{\mathrm{critic}} = 5, m = 40, l = 10000n, \alpha = 0.00001, \beta_1 = 0.5, \beta_2 = 0.9$

---
$P_0 \longleftarrow \mathbf{0}, \theta_0 \longleftarrow \mathbf{0}$
**for** $t = 1, \cdots, T$ **do**
    Minimax realization of $\min_\theta W(\phi_\theta, \mu_{\mathrm{emp}}) + \lambda T(\theta)$ **(*)**:
    **while** $\theta$ has not converged **do**
        **for** $s = 1, ..., n_{\mathrm{critic}}$ **do**
            Discriminator updates $w$
        Sample $\{\mathbf{z}_i\}_{i=1}^m, \{\mathbf{z}_i'\}_{i=1}^m \sim p(\mathbf{z})$
        $L \longleftarrow -\frac{1}{m} \sum_{i=1}^m f_w(g_\theta(\mathbf{z}_i)) + \lambda \frac{\sum_{i,j} \Xi(\theta, \mathbf{z}_i, \mathbf{z}_j')}{m^2}$ ($\Xi$ is defined in equation 25)
        $\theta \leftarrow \mathrm{Adam}(\nabla_\theta L, \theta, \alpha, \beta_1, \beta_2)$
    $\theta_t \longleftarrow \theta$
    Realization of step **(**):
    Sample $\{\mathbf{z}_i\}_{i=1}^l, \{\mathbf{z}_i'\}_{i=1}^l \sim p(\mathbf{z})$
    $M_t \longleftarrow \sum_{ij} g_{\theta_t}(\mathbf{z}_i) g_{\theta_t}(\mathbf{z}_j')^T M(g_{\theta_t}(\mathbf{z}_i), g_{\theta_t}(\mathbf{z}_j'))$
    Find $\{\mathbf{v}_i\}_1^n$ s.t. $M_t \mathbf{v}_i = \lambda_i \mathbf{v}_i, \lambda_1 \geq \cdots \geq \lambda_n$
    $P_t \longleftarrow \sum_{i=1}^k \mathbf{v}_i \mathbf{v}_i^T$
**Output:** $\mathbf{v}_1, \cdots, \mathbf{v}_k$

---

## I.2 How to estimate $\frac{\partial T}{\partial \theta}$ and $M_{\phi_{\theta_t}}$?

Another important aspect of the numerical algorithm is the complexity of estimating the matrix $M_{\phi_{\theta_t}}$ at step **(**). The following theorem shows that we only need to sample $\mathbf{z} \sim p$ a sufficient number of times to estimate $\frac{\partial T}{\partial \theta}$ and $M_{\phi_{\theta_t}}$.

**Theorem 14.** *If $\phi_\theta$ is pdf of the random vector $g_\theta(\mathbf{z})$, $\mathbf{z} \sim p(\mathbf{z})$, then*

$$\frac{\partial T}{\partial \theta} = \mathbb{E}_{\mathbf{z}, \mathbf{z}' \sim p} \frac{\partial \Xi(\theta, \mathbf{z}, \mathbf{z}')}{\partial \theta}$$

$$M_{\phi\theta} = \mathbb{E}_{\mathbf{z}, \mathbf{z}' \sim p} g_\theta(\mathbf{z}) g_\theta(\mathbf{z}')^T M(g_\theta(\mathbf{z}), g_\theta(\mathbf{z}'))$$

*where*

$$\begin{aligned}
\Xi(\theta, \mathbf{z}, \mathbf{z}') = (g_\theta(\mathbf{z}) \cdot g_\theta(\mathbf{z}')) M(g_\theta(\mathbf{z}), g_\theta(\mathbf{z}')) - \\
2(g_\theta(\mathbf{z}) \cdot P_{t-1} g_{\theta_{t-1}}(\mathbf{z}')) M(g_\theta(\mathbf{z}), g_{\theta_{t-1}}(\mathbf{z}'))
\end{aligned} \tag{25}$$

*and RHS is well-defined.*

### I.2.1 DEFINITION OF $\mathcal{H}$

Specifically, for robust PCA/outlier pursuit applications, we define $\phi_\theta(\mathbf{x})$ as a probability density function of the random vector $\mathbf{a} + \mathbf{b}$, where $\mathbf{a}, \mathbf{b}$ are independent and $\mathbf{a}$ is the $i$-th column of matrix $\theta_1 \in \mathbb{R}^{n \times N}$ (where $i \sim \mathcal{U}(1, N)$ is sampled uniformly from $\{1, \cdots, N\}$), $\mathbf{b} = g_{\theta_2}(\mathbf{c})$, $\mathbf{c} \sim \mathcal{N}(\mathbf{0}, I_n)$ and $g_{\theta_2} : \mathbb{R}^n \rightarrow \mathbb{R}^n$ is a neural network with weights $\theta_2$. Thus, $\theta = (\theta_1, \theta_2)$. It can be checked that $\mathcal{H}$, defined in this way, satisfies the Assumption 1. We specifically introduce the random vector $\mathbf{a}$ here because, according to Theorem 6, the ultimate solution of the problem corresponds to $\theta_1 = Y$ and $\mathbf{b} = \mathbf{0}$. This guarantees that the solution is approachable from set $\mathcal{H}$.

### I.3 PROOF OF THEOREM 14

We need to following lemma.

**Lemma 5.** $||S_\phi - PS_\psi||^2 = \mathbb{E}_{\mathbf{x}, \mathbf{y} \sim \phi}(\mathbf{x} \cdot \mathbf{y}) M(\mathbf{x}, \mathbf{y}) + \mathbb{E}_{\mathbf{x}, \mathbf{y} \sim \psi}(\mathbf{x} \cdot P\mathbf{y}) M(\mathbf{x}, \mathbf{y}) - 2\mathbb{E}_{\mathbf{x} \sim \phi, \mathbf{y} \sim \psi}(\mathbf{x} \cdot P\mathbf{y}) M(\mathbf{x}, \mathbf{y})$

*Proof of lemma.*

$$||S_\phi - PS_\psi||^2 = ||\sqrt{O(M)}[\mathbf{x}\phi(\mathbf{x})] - P\sqrt{O(M)}[\mathbf{x}\psi(\mathbf{x})]||^2 =$$

$$= ||\sqrt{O(M)}[\mathbf{x}\phi(\mathbf{x}) - P\mathbf{x}\psi(\mathbf{x})]||^2 = \sum_{i=1}^{n} ||\sqrt{O(M)}[x_i \phi(\mathbf{x}) - (P\mathbf{x})_i \psi(\mathbf{x})]||^2 =$$

$$\sum_{i=1}^{n} \langle x_i \phi(\mathbf{x}) | O(M)[x_i \phi(\mathbf{x})] \rangle + \langle (P\mathbf{x})_i \psi(\mathbf{x}) | O(M)[(P\mathbf{x})_i \psi(\mathbf{x})] \rangle - 2\langle (P\mathbf{x})_i \psi(\mathbf{x}) | O(M)[x_i \phi(\mathbf{x})] \rangle =$$

$$\mathbb{E}_{\mathbf{x}, \mathbf{y} \sim \phi}(\mathbf{x} \cdot \mathbf{y}) M(\mathbf{x}, \mathbf{y}) + \mathbb{E}_{\mathbf{x}, \mathbf{y} \sim \psi}(\mathbf{x} \cdot P\mathbf{y}) M(\mathbf{x}, \mathbf{y}) - 2\mathbb{E}_{\mathbf{x} \sim \phi, \mathbf{y} \sim \psi}(\mathbf{x} \cdot P\mathbf{y}) M(\mathbf{x}, \mathbf{y})$$

$\square$

*Proof of theorem 14.* Using lemma 5 we have:

$$T(\theta) = \mathbb{E}_{\mathbf{x}, \mathbf{y} \sim \phi_\theta}(\mathbf{x} \cdot \mathbf{y}) M(\mathbf{x}, \mathbf{y}) + \mathbb{E}_{\mathbf{x}, \mathbf{y} \sim \phi_{\theta_{t-1}}}(\mathbf{x} \cdot P_{t-1}\mathbf{y}) M(\mathbf{x}, \mathbf{y}) -$$

$$-2\mathbb{E}_{\mathbf{x} \sim \phi_\theta, \mathbf{y} \sim \phi_{\theta_{t-1}}}(\mathbf{x} \cdot P_{t-1}\mathbf{y}) M(\mathbf{x}, \mathbf{y}) =$$

$$\mathbb{E}_{\mathbf{z}, \mathbf{z}' \sim p}(g_\theta(\mathbf{z}) \cdot g_\theta(\mathbf{z}')) M(g_\theta(\mathbf{z}), g_\theta(\mathbf{z}')) +$$

$$\mathbb{E}_{\mathbf{z}, \mathbf{z}' \sim p}(g_{\theta_{t-1}}(\mathbf{z}) \cdot P_{t-1}g_{\theta_{t-1}}(\mathbf{z}')) M(g_{\theta_{t-1}}(\mathbf{z}), g_{\theta_{t-1}}(\mathbf{z}')) -$$

$$2\mathbb{E}_{\mathbf{z}, \mathbf{z}' \sim p}(g_\theta(\mathbf{z}) \cdot P_{t-1}g_{\theta_{t-1}}(\mathbf{z}')) M(g_\theta(\mathbf{z}), g_{\theta_{t-1}}(\mathbf{z}'))$$

The second term does not depend on $\theta$. Therefore,

$$\frac{\partial T}{\partial \theta} = \frac{\partial}{\partial \theta} \mathbb{E}_{\mathbf{z}, \mathbf{z}' \sim p} \Xi(\theta, \mathbf{z}, \mathbf{z}')$$

where

$$\Xi(\theta, \mathbf{z}, \mathbf{z}') = (g_\theta(\mathbf{z}) \cdot g_\theta(\mathbf{z}')) M(g_\theta(\mathbf{z}), g_\theta(\mathbf{z}')) - 2(g_\theta(\mathbf{z}) \cdot P_{t-1}g_{\theta_{t-1}}(\mathbf{z}')) M(g_\theta(\mathbf{z}), g_{\theta_{t-1}}(\mathbf{z}'))$$

If $\mathbb{E}_{\mathbf{z},\mathbf{z}'\sim p}\frac{\partial\Xi(\theta,\mathbf{z},\mathbf{z}')}{\partial\theta}$ is well-defined (the proof of sufficiency of that condition is similar to the proof of Theorem 3 from Arjovsky et al. (2017)), then, using Leibniz integral rule, we obtain:

$$\frac{\partial}{\partial\theta}\mathbb{E}_{\mathbf{z},\mathbf{z}'\sim p}\Xi(\theta,\mathbf{z},\mathbf{z}')=\mathbb{E}_{\mathbf{z},\mathbf{z}'\sim p}\frac{\partial\Xi(\theta,\mathbf{z},\mathbf{z}')}{\partial\theta}$$

The fact that

$$M_{\phi_\theta}=\mathbb{E}_{\mathbf{z},\mathbf{z}'\sim p}g_\theta(\mathbf{z})g_\theta(\mathbf{z}')^T M(g_\theta(\mathbf{z}),g_\theta(\mathbf{z}'))$$

is obvious from the definition $M_{\phi_\theta}=\mathbb{E}_{\mathbf{x},\mathbf{y}\sim\phi_\theta}\mathbf{x}\mathbf{y}^T M(\mathbf{x},\mathbf{y})$. $\qquad\square$

## J  A NUMERICAL ALTERNATING SCHEME FOR SDR

For a binary classification case, given a labeled dataset $\{(\mathbf{x}_i,y_i)\}_{i=1}^N$, $\mathbf{x}_i\in\mathbb{R}^n$, $y_i\in\mathcal{C}$, $\mathcal{C}=\{0,1\}$ we formulate the sufficient dimension reduction problem as the minimization task:

$$J(f)=\mathbb{E}_{(\mathbf{z},c)\sim\mu_{\text{data}},\boldsymbol{\epsilon}\sim N(\mathbf{0},\upsilon^2 I_n)}L(c,f(\mathbf{z}+\boldsymbol{\epsilon}))\to\min_{f\in\mathcal{F}_k}$$

where $L(c,y)=-c\log(y)-(1-c)\log(1-y)$.

We apply the alternating scheme in the dual space (Algorithm 2) to this task. We set $M(\mathbf{x},\mathbf{y})=\zeta(\mathbf{x}-\mathbf{y})$, where $\hat{\zeta}$ is a strictly positive probability density function. A numerical version of the scheme is given below (Algorithm 8).

At every iteration $t=1,\cdots,T$ of the Algorithm 2 we solve the task (in our case $\tilde{I}=J$):

$$\hat{\phi}_t\leftarrow\arg\min_{\hat{\phi}}\tilde{I}(\hat{\phi})+\tilde{\lambda}||\,|\frac{\partial\hat{\phi}}{\partial\mathbf{x}}-P_{t-1}\frac{\partial\hat{\phi}_{t-1}}{\partial\mathbf{x}}|\,||^2_{L_{2,\hat{\zeta}}(\mathbb{R}^n)}$$

In a numerical version of the algorithm we assume that $\hat{\phi}$ is given as a neural network $f_\theta$, i.e. our task becomes:

$$\theta_t\leftarrow\arg\min_\theta J(f_\theta)+\tilde{\lambda}\mathbb{E}_{\boldsymbol{\xi}\sim\hat{\zeta}}||\frac{\partial f_\theta}{\partial\mathbf{x}}(\boldsymbol{\xi})-P_{t-1}\frac{\partial f_{\theta_{t-1}}}{\partial\mathbf{x}}(\boldsymbol{\xi})||^2$$

The gradient of the function $\Phi(\theta)=J(f_\theta)+\tilde{\lambda}\mathbb{E}_{\boldsymbol{\xi}\sim\hat{\zeta}}||\frac{\partial f_\theta}{\partial\mathbf{x}}(\boldsymbol{\xi})-P_{t-1}\frac{\partial f_{\theta_{t-1}}}{\partial\mathbf{x}}(\boldsymbol{\xi})||^2$ equals:

$$\frac{\partial\Phi(\theta)}{\partial\theta}=\mathbb{E}_{(\mathbf{z},c)\sim P_{\text{data}},\boldsymbol{\epsilon}\sim N(\mathbf{0},\upsilon^2 I_n)}\frac{\partial}{\partial\theta}L(c,f_\theta(\mathbf{z}+\boldsymbol{\epsilon}))+\tilde{\lambda}\mathbb{E}_{\boldsymbol{\xi}\sim\hat{\zeta}}\frac{\partial}{\partial\theta}||\frac{\partial f_\theta}{\partial\mathbf{x}}(\boldsymbol{\xi})-P_{t-1}\frac{\partial f_{\theta_{t-1}}}{\partial\mathbf{x}}(\boldsymbol{\xi})||^2$$

That is why $\nabla_\theta L$ (given to Adam optimizer in the gradient descent loop) in the Algorithm 8 is an unbiased estimator of $\frac{\partial\Phi(\theta)}{\partial\theta}$. Thus, in the "while loop" we find optimal $\hat{\phi}_t=f_{\theta_t}$.

According to Algorithm 2, the next goal is to estimate $M_t=\left[\text{Re}\,\langle\frac{\partial\hat{\phi}_t}{\partial x_i},\frac{\partial\hat{\phi}_t}{\partial x_j}\rangle_{L_{2,\hat{\zeta}}(\mathbb{R}^n)}\right]$. It is easy to see that

$$M_t=\mathbb{E}_{\boldsymbol{\chi}\sim\hat{\zeta}}\frac{\partial\hat{\phi}_t}{\partial\mathbf{x}}(\boldsymbol{\chi})\frac{\partial\hat{\phi}_t}{\partial\mathbf{x}}(\boldsymbol{\chi})^T=\mathbb{E}_{\boldsymbol{\chi}\sim\hat{\zeta}}\frac{\partial f_{\theta_t}}{\partial\mathbf{x}}(\boldsymbol{\chi})\frac{\partial f_{\theta_t}}{\partial\mathbf{x}}(\boldsymbol{\chi})^T$$

From the last we see that the matrix $M_t$ can be estimated by sampling $\boldsymbol{\chi}\sim\hat{\zeta}$ a sufficient number of times (the parameter $l$ in our algorithm). All the rest is identical to Algorithm 2.

The regression version of the algorithm can be obtained by setting $L(c,c')=(c-c')^2$. Implementations for different databases can be found at github.

**Algorithm 8** The numerical alternating scheme for SDR. We use $\upsilon = 1.0, \hat{\zeta}(\mathbf{x}) = G_{0.8}^n(\mathbf{x})$ and default values of $\tilde{\lambda} = 10, m \approx 50, m' = 100, l = 30000, \alpha = 0.0001, \beta_1 = 0.5, \beta_2 = 0.9$

---

$P_0 \longleftarrow \mathbf{0}, \theta_0 \longleftarrow \mathbf{0}$
**for** $t = 1, \cdots, T$ **do**
    **while** $\theta$ has not converged **do**
        Sample $\{(\mathbf{z}_i, c_i)\}_{i=1}^m \sim P_{\text{data}}$
        Sample $\{\boldsymbol{\epsilon}_i\}_{i=1}^m \sim N(\mathbf{0}, \upsilon^2 I_n)$
        Sample $\{\boldsymbol{\xi}_i\}_{i=1}^{m'} \sim \hat{\zeta}$
        $L \longleftarrow \frac{1}{m} \sum_{i=1}^m L(c_i, f_\theta(\mathbf{z}_i + \boldsymbol{\epsilon}_i)) + \frac{\tilde{\lambda}}{m'} \sum_{i=1}^{m'} ||\frac{\partial f_\theta(\boldsymbol{\xi}_i))}{\partial \mathbf{x}} - P_{t-1} \frac{\partial f_{\theta_{t-1}}(\boldsymbol{\xi}_i))}{\partial \mathbf{x}}||^2$
        $\theta \longleftarrow \text{Adam}(\nabla_\theta L, \theta, \alpha, \beta_1, \beta_2)$
    $\theta_t \longleftarrow \theta$
    Sample $\{\boldsymbol{\chi}_i\}_{i=1}^l \sim \hat{\zeta}$
    Calculate $M_t = \frac{1}{l} \sum_{i=1}^l \frac{\partial f_{\theta_t}(\boldsymbol{\chi}_i))}{\partial \mathbf{x}} \frac{\partial f_{\theta_t}(\boldsymbol{\chi}_i))}{\partial \mathbf{x}}^T$
    Find $\{\mathbf{v}_i\}_1^n$ s.t. $M_t \mathbf{v}_i = \lambda_i \mathbf{v}_i, \lambda_1 \geq \cdots \geq \lambda_n$
    $P_t \longleftarrow \sum_{i=1}^k \mathbf{v}_i \mathbf{v}_i^T$
**Output:** $\mathbf{v}_1, \cdots, \mathbf{v}_k$

---

