# OpenReview forum: "Dimension reduction as an optimization problem over a set of generalized functions"
_ICLR.cc/2021/Conference — Reject_

### Official Review · AnonReviewer2 · 2020-10-24
**Generally not written in a widely accessible manner**

**Rating:** 5
**Confidence:** 2

**Review:**

Summary: In the dimension reduction problem, we are given a set of high-dimensional points and would like to embed them in a lower dimensional space so as to preserve relevant properties. This paper proposes a certain optimization framework for dimension reduction, and suggests to solve it by alternating optimization. The problem is formulated in terms of a general distance measure between distributions, and the alternating optimization scheme requires instantiating a certain optimization step for the specific choice of distance measure. The paper instantiates it for four such choices. The method is implemented and some experiments are presented.

Comments:
Much of the paper consists of highly technical mathematical derivations, and unfortunately I do not have the background to assess this content. As a matter of presentation, it seems to me an unfortunate choice to not even define the basic notions on which the paper is based (Schwartz space, tempered distributions, generalized functions etc); instead, the paper just refers to textbooks. I do not believe the current manuscript is widely accessible to ICLR audience, as it seems to require a rather specialized background in functional analysis. An introductory section or appendix defining the basic terms and facts is standard in such cases, and would go a long way in making the paper accessible and self-contained.

The experiments do not show a clear advantage of the proposed methods; in fact, when compared with two other baselines, each of the three methods gets the best result on exactly one third of the experiments in Table 1, and the differences between them are generally small and doubtedly significant. Is there a further message in these results that point to some advantages of the proposed method?

The real datasets used are mentioned by name and with a link to their UCI source. This seems insufficient; please include a written out reference or link for them, as hidden hyperlinks are not visible and are of course lost when the paper is printed. It would also help to specify the parameters of the datasets (number of points, ambient dimension etc), which are needed in order to put your empirical results in context.

I could not find a discussion of the running time / scalability of the proposed method compared to PCA and the other baselines. It would help if the authors could comment on that (both asymptotically and in practice) as it is relevant to assessing their claim that their method is suitable for use instead of PCA.

As a final remark on presentation, please use the \citep command where appropriate. Currently most of the references are unbracketed and make the text difficult to read (e,g., "the problem becomes of special interest Cunningham & Ghahramani (2015)", and many other such instances).

Conclusion: The mathematical content of the manuscript is largely opaque to me. The experimental results are not outstanding, though perhaps the new approach presented here has some conceptual merit that could justify acceptance (this is currently difficult for me to judge). Given the diverse spectrum of ICLR target audiences, I would advise revising the presentation to be friendlier to the general ML community.

Post-rebuttal update: I thank the authors for their response. The points raised above were largely acknowledged and the authors chose to not revise the manuscript, so my assessment remains the same.

---

### Official Review · AnonReviewer1 · 2020-10-25
**Clean theory**

**Rating:** 7
**Confidence:** 3

**Review:**

The paper considers the unsupervised dimension reduction problem, in which one is given a finite number of points in R^n drawn from some distribution and wants to find a low-dimensional distribution that approximates the underlying unknown distribution. The paper proposes to solve a minimization problem min_f { I(f) + lambda*R(f) }, where I(f) is the distance between f and the empirical distribution of the data and R(f) is a penalty term that tries to force f to be low-dimensional. When the penalty is defined via a kernel function, R(f) admits a highly tractable form and, for a given f, can be solved by an SVD. This leads to an alternating scheme to minimize I(f) + lambda*R(f), which is the main contribution of the paper. The algorithm is iterative. In each iteration, it finds the minimizer f with respect to the distance to the empirical distribution and the penalty in terms of a k-dimensional subspace found in the previous iteration, then finds a new k-dimensional space by minimizing the penalty. The paper then conducts experiments by applying this general theory to specific distance functions and kernels.

Strengths:
- Theory stated in general normed spaces, clean and neat
- Algorithm is conceptually clear and simple
- Experiments seem to confirm that the proposed alternating scheme is a serious competitor with the existing Sliced Inverse Regression and Kernel Dimensionality Reduction algorithms.

Weaknesses:
- It is not clear how one should choose a finite-dimensional domain of phi (which is defined in an infinite-dimensional space). Although the authors have specified some domains for specific problems in the Appendix, in general, it is not clear how to choose the domain phi and how to solve the optimization problem in Line 3 of Algorithm 1. It is also not clear when to use the primal and when to use the dual form of Algorithm 1. Some discussions are expected to be included in the main body.
- No convergence analysis of Algorithm 1?
- It would be better to compare the running time with the existing algorithms, too.

Minor points:
- page 2, end of Section 2, “Identity matrix” -> “The identity matrix”
- page 5, two lines above Theorem 4: “a real part” -> “the real part”
- page 5 and 6, “task 11”, “problem 11”, etc -> “11” should be “(11)” so that readers know it refers to Equation (11).
- page 6, the paragraph below Algorithm 1, “Scheme 1” -> “Algorithm 1”?

---

### Official Review · AnonReviewer4 · 2020-10-30
**Reformulation of unsupervised dimensionality reduction problem**

**Rating:** 4
**Confidence:** 4

**Review:**

The paper considers the unsupervised dimension reduction problem. That is, given a set of points in R^n, find a low dimensional affine subspace that approximate the support of the distribution that generated the points. More specifically, the paper considers the empirical probability density function p_emp of a dataset which is the average of \delta^n(x-x_i), where x_i's are points of the dataset and \delta^n is the n-dimensional version of the Dirac function. Then the goal is to find a distribution q such that its density is supported in a k-dimensional affine space and it minimized a certain loss D(p_emp,q), where D is a measure of distance between two distributions.
The paper then presents 4 examples of problems that can be formulated in this framework: 1) maximum mean discrepancy; 2) distance based on the higher moments; 3) Wasserstein distance; and 4) sufficient dimension reduction.
Finally, the paper proposes an alternating optimization scheme to solve this optimization problem and presents experiments that compare the accuracy of the proposed method with other dimensionality reduction methods like PCA.

I think the paper is not well-motivated, and it is not clear what are the novelties of the paper. Please explicitly state what are the contributions of the paper. The experiments are also very inconclusive. Table 1 reports the accuracy of KNN on 2 and 3 dimensions. First, I think it is better to report the reconstruction error of PCA and other methods instead of this. Moreover, it is better to test the projection on a bit higher dimensions as well. For example what happens for k=10?

---

### Author Response · Authors · 2020-11-15
**We thank reviewers for reviews**

We thank reviewers for reviews and appreciating our work. We clarify the main questions below,

AnonReviewer1: Yes, we intentionally did not indicate the domain over which we minimize in line 3 of the general scheme Alg.1. The only thing that matters is that this domain must be "rich enough" to capture solutions of (11). In practice, for the SDR case, we define the domain as feed-forward neural networks (that we know to approximate all interesting functions). For the Wasserstein case, we define the domain as a set of probability distributions of the random vector G(z) where z is the multivariate Gaussian, and G is a feed-forward neural network (it is known that such distributions capture all interesting distributions). In the MMD and the HM cases, we exploit the dual form of the alternating scheme; that is why we define the domain according to equation (20) (it is in the appendix part).

When to use the primal part and when to use the dual? Our computational experiments show that the dual algorithm is faster than the primal one (though we have not compared the primal and the dual algorithms efficiencies for the same objective yet).
We believe that this is mainly because we used the primal version only for the Wasserstein case, where line 3 of the general scheme Alg.1 is implemented as the minimax GAN-style algorithm (which is computationally heavier than straightforward gradient descent that we use in the dual algorithm).

We believe that potentially both primal and dual versions of the alternating scheme can be turned into efficient practical algorithms. E.g., in the paper, we give the dual algorithm for the MMD case, though the primal algorithm is also easy to implement, using ideas for gradient computation from (Li & Swersky & Zemel, 2015). This implementation and the convergence analysis of algorithm 1 is planned as the future work.

AnonReviewer1 & AnonReviewer2: If to compare the running times of our MMD/HM/Wasserstein with PCA (for the dimensionality reduction case), of our SDR with KDR/SIR (for the sufficient dimension reduction case), then our algorithms are much slower. The main advantage that we claim of using the alternating scheme is not its speed, but the possibility to optimize new kinds of objectives. Thus, we obtain an optimization tool for a new family of objectives.

AnonReviewer2: Yes, the content of the paper indeed requires some knowledge of tempered distributions. We tried to make the narrative as self-contained as possible, but eight pages are too small to put definitions of distributions theory into the main part of the paper. In the final version, we can put some definitions into the Appendix part.

AnonReviewer4: In Table 1 we compare our SDR algorithm with SIR/KDR. In SDR literature, using k=2,3 is very common, and, instead of the reconstruction error, the Nearest Neighbour classification error is computed on reduced datasets (e.g., see Yamada & Niu & Takagi & Sugiyama, 2011). As k grows, the classification accuracy/R^2 on the test set monotonically improves but only slightly, and achieves a plateau. In our experiments, the speed of the SDR algorithm does not depend on k.

General comment on the main contribution: Generalized functions are already exploited in such contexts as, e.g., kernel density estimation, signal processing, etc. But to our knowledge, this is the first attempt to apply them to the dimensionality reduction problem. Distributions appear in that context very naturally and in a full manner. I.e., using this language leads to a general view of DR and SDR as one single topic. Avoiding using this language (i.e., using only probability measures and functions independently in every specific case) leads to unnecessary narrowing of the view. Also, when we avoid using this language, we cannot use the Fourier transform theory in its full capacity.

---

### Decision · Program_Chairs · 2021-01-07
**Final Decision**

**Decision:**

Reject

**Comment:**

Overall, there were significant concerns about the motivation and experiments in this paper, and these were thought not to merit acceptance on their own. Because of this, the reviewers started discussing the theory to see if that would justify acceptance. The reviewers were not able to find a clear advantage over existing approaches, nor sufficient motivation; also the presentation was found to be largely inaccessible. In the rebuttal there was a brief mentioning of background and possible implications, but they were hard to assess and the paper itself did not have such context nor was updated to have such context. For a future version, one recommendation could be to focus significantly more on context, motivation, and improvements over prior work. Also, making the paper more self-contained could help.